# Phase transitions in when feedback is useful

**Lokesh Boominathan**
Department of ECE
Rice University
Houston, TX 77005
lokesh.boominathan@rice.edu

**Xaq Pitkow**
Dept. of Neuroscience, Dept. of ECE
Baylor College of Medicine, Rice University
Houston, TX 77005
xaq@rice.edu

## Abstract

Sensory observations about the world are invariably ambiguous. Inference about the world's latent variables is thus an important computation for the brain. However, computational constraints limit the performance of these computations. These constraints include energetic costs for neural activity and noise on every channel. Efficient coding is one prominent theory that describes how such limited resources can best be used. In one incarnation, this leads to a theory of predictive coding, where predictions are subtracted from signals, reducing the cost of sending something that is already known. This theory does not, however, account for the costs or noise associated with those predictions. Here we offer a theory that accounts for both feedforward and feedback costs, and noise in all computations. We formulate this inference problem as message-passing on a graph whereby feedback serves as an internal control signal aiming to maximize how well an inference tracks a target state while minimizing the costs of computation. We apply this novel formulation of *inference as control* to the canonical problem of inferring the hidden scalar state of a linear dynamical system with Gaussian variability. The best solution depends on architectural constraints, such as Dale's law, the ubiquitous law that each neuron makes solely excitatory or inhibitory postsynaptic connections. This biological structure can create asymmetric costs for feedforward and feedback channels. Under such conditions, our theory predicts the gain of optimal predictive feedback and how it is incorporated into the inference computation. We show that there is a non-monotonic dependence of optimal feedback gain as a function of both the computational parameters and the world dynamics, leading to phase transitions in whether feedback provides any utility in optimal inference under computational constraints.

## 1 Introduction

A critical computation for the brain is to infer the world's latent variables from ambiguous observations. Computational constraints, including metabolic costs and noisy signals, limit the performance of these inferences. Efficient coding [1] is a prominent theory that describes how limited resources can be used best. In one incarnation, this leads to the theory of predictive coding [2], which posits that predictions are sent along feedback channels to be subtracted from signals at lower cortical areas; only the difference returns to the higher areas along feedforward channels, reducing the metabolic or informational cost of sending redundant signals already known to the higher areas. This theory does not, however, account for the additional costs or noise associated with the feedback. Depending on the costs for sending predictions and the reliability of signals encoding those predictions, we expect different optimal strategies to perform computationally constrained inferences. For example, if the feedback channel is too unreliable and expensive, we hypothesize that it is not worth sending any predictions at all. Here we offer a more general theory of inference that accounts for the costs and reliabilities of the feedback and feedforward channels, and the relative importance of good inferences

36th Conference on Neural Information Processing Systems (NeurIPS 2022).

about the latent world state. We formulate the inference problem as control via message-passing on a graph, maximizing how well an inference tracks a target state while minimizing the message costs. Messages become control actions with their own costs to reduce while improving how well an inference tracks a target state. We call this method *inference as control*, as it flips the interesting perspective of viewing optimal control as an inference problem [3]. We solve this problem under Linear-Quadratic-Gaussian (LQG) assumptions: Linear dynamics, Quadratic state and control costs, and Gaussian noise for the process, observations, and messages. Our theory enables us to determine the optimal predictions and how they are integrated into computationally constrained inference. This analysis reveals phase transitions in when feedback is helpful, as we change the computation parameters or the world dynamics. Finally, we connect our theory's constraints to biology by providing a simplified example of how feedback/feedforward pathways with different anatomical structures can yield different optimal strategies.

## 2    Related work

Our work brings together several related theories: Bayesian inference, efficient coding, and predictive coding. The idea that the brain performs Bayesian inference about latent variables amid uncertain sensory observation has been long studied in neuroscience [4–7]. Bayesian inference involves optimally combining prior information from the past or from surrounding context with the likelihood of current sensory observations [8–11]. The theory of efficient coding [1, 12–15] focuses on encoding the sensory observations themselves, capturing maximal information subject to limited biological resources. The theory of predictive coding [2] tackles such resource constraints by using top-down feedback predictions to suppress the part of sensory observations that is already known to the brain, and sending only the novel part of the observation back to the brain [16–21]. Understanding the utility of predictive feedback has also shown promise in gaining insights into deep learning algorithms and improving their performance, recently drawing wide interest in this topic [22–29].

Park *et al.* [30] brought together the ideas of a Bayesian brain and efficient coding, by identifying efficient neural codes as optimizing a posterior distribution while accounting for limited firing rates. Chalk *et al.* [18] unified theories of predictive coding, efficient coding, and sparse coding by showing how these regimes emerge in a three dimensional constraint space described by channel capacity, past information content about the world state, and the time point to which the estimate is targeted. Młynarski *et al.* [31] investigated how sensory coding can adapt over time to account for the trade-off of an inference cost against a metabolic cost for high-fidelity encoding. Aitchison *et al.* [32] argued that Bayesian inference can be achieved using predictive coding, but is not necessary. While these works made important contributions towards unifying different encoding schemes that are optimal under different circumstances, they all assume that information is costly but computation is free. In particular, none of them have explicitly accounted for biological constraints along the feedback channel as well. In our work, we discuss how the optimal strategy changes when balancing inference performance against energetic costs, when there is noise in both feedforward and feedback pathways.

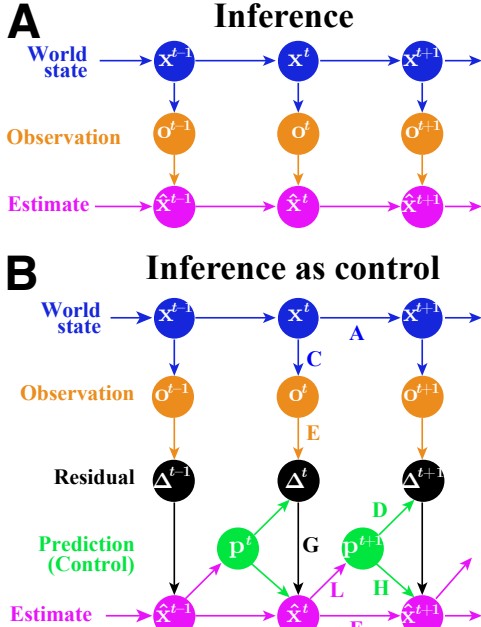

Figure 1: *Resource constrained inference modeled as a control problem.* **A**: *Graphical model of the inference problem, tracking a latent state in a Hidden Markov Model.* **B**: *Expanded inference problem, indicating prediction as control.*

## 3    Defining the problem

We consider an inference task in which the brain tracks a latent world state $x^t$ based on its noisy sensory observations $o^t$ (Fig 1A). The trajectory of the world state follows a known stochastic linear

dynamical system (Eq 1) with Gaussian process noise and Gaussian observations (Eq 2). At each time step, the brain sends a top-down prediction $p^t$ based on its best estimate $\hat{x}^{t-1}$ based on evidence up until the previous time step (Eq 3). The prediction is then sent through an additive Gaussian noise feedback channel to the sensory level (Eq 4). The noisy prediction $\tilde{p}^t$ is then combined with the new observations $o^t$ to form a residual $\Delta^t$ (Eq 5). The residual is then sent through an additive Gaussian noise feedforward channel to the brain (Eq 6). Based on this noisy residual $\tilde{\Delta}^t$ and the prediction it had just sent, the brain updates its estimate $\hat{x}^t$ (Eq 7). Fig 1B shows a trellis diagram of the dynamics.

Structurally, these dynamics are equivalent to a Kalman filter. The most common representation of this filter might even be viewed as predictive coding, as the update step uses the residual between the predicted and actual observation. However, the Kalman filter has no costs aside from the final inference, and no computational noise except the observation. Biological computation therefore may weigh fundamentally different tradeoffs in its inferences. Our goal is to find the parameters that minimize a weighted combination of inference loss, feedback energy cost, and feedforward energy cost (Eq 8) given limitations caused by computational noise.

We optimize the following parameters: the gain $L$ on the previous inference that is integrated into the new prediction; the prediction multiplier $D$ and observation multiplier $E$ that describe how the noisy prediction and the observation are weighted to form the residual; and the parameters $F$, $G$, and $H$ that determine how the inference is updated in light of the new noisy residual. The optimization is done at steady state, assuming the observer must continually update its estimate in a stationary dynamic environment. For mathematical tractability, we assume that all messages are linear functions of their inputs, the three losses are quadratic, the weight on the inference loss is a scalar, and all noise is independent Gaussian white noise. The equations governing this problem are:

$$x^t = A\,x^{t-1} + \eta_{\mathrm{p}}^t \qquad\qquad \text{state dynamics} \qquad (1)$$

$$o^t = C\,x^t + \eta_{\mathrm{o}}^t \qquad\qquad \text{observation} \qquad (2)$$

$$p^t = L\,\hat{x}^{t-1} \qquad\qquad \text{prediction} \qquad (3)$$

$$\tilde{p}^t = p^t + \eta_{\mathrm{b}}^t \qquad\qquad \text{noisy prediction, feedback} \qquad (4)$$

$$\Delta^t = D\,\tilde{p}^t + E\,o^t \qquad\qquad \text{residual} \qquad (5)$$

$$\tilde{\Delta}^t = \Delta^t + \eta_{\mathrm{f}}^t \qquad\qquad \text{noisy residual, feedforward} \qquad (6)$$

$$\hat{x}^t = F\,\hat{x}^{t-1} + G\,\tilde{\Delta}^t + H\,p^t \qquad\qquad \text{estimation} \qquad (7)$$

$$\mathrm{Cost}_{\mathrm{tot}} = \lim_{T\to\infty}\frac{1}{T}\sum\nolimits_{t=1}^{T}\bigg(\underbrace{\|x^t - \hat{x}^t\|^2}_{\mathrm{Cost_{inf}}} + \underbrace{\tilde{p}^{t\top} W_{\mathrm{b}}\,\tilde{p}^t}_{\mathrm{Cost_b}} + \underbrace{\tilde{\Delta}^{t\top} W_{\mathrm{f}}\,\tilde{\Delta}^t}_{\mathrm{Cost_f}}\bigg) \qquad (8)$$

We consider how the optimal computational strategy varies with cost-weights $(W_{\mathrm{b}}, W_{\mathrm{f}})$ that determine the relative importance of feed**b**ack, feed**f**orward, and **inf**erence costs ($\mathrm{Cost_b}$, $\mathrm{Cost_f}$, and $\mathrm{Cost_{inf}}$). $\eta_{\mathrm{p}}, \eta_{\mathrm{o}}, \eta_{\mathrm{b}},$ and $\eta_{\mathrm{f}}$ represent the process noise, observation noise, feedback noise, and feedforward noise with variances $\sigma_{\mathrm{p}}^2, \sigma_{\mathrm{o}}^2, \sigma_{\mathrm{b}}^2,$ and $\sigma_{\mathrm{f}}^2$ respectively. We will see below that certain combinations of parameters determine the system's behavior at transition points.

## 4    Method: Inference as Control

We adopt a two-step approach to solve the optimization problem in Eq 8. First we fix the parameters $D$ and $E$ that determine how the feedback is used to subtract predictions, and solve in closed form for the optimal feedback gain ($L$) and optimal integration of residuals ($F$, $G$, $H$) as a function of the fixed parameters. We then numerically optimize for $D$ and $E$, given the optimal feedback. Mathematically, we write the two-step minimization as below, with an $\mathrm{argmin}$ determined analogously.

$$\min_{D,E,L,F,G,H} \mathrm{Cost}_{\mathrm{tot}} = \min_{D,E}\left(\min_{L,F,G,H} \mathrm{Cost}_{\mathrm{tot}}\right) \qquad (9)$$

In order to find the closed form solution for fixed $D$ and $E$, we recast the minimization as an LQG control problem [33] where the prediction is treated as an internal control. The LQG dynamical Eqs 10-11 are obtained by concisely writing Eqs 1-2, and 4-6 in terms of an augmented state $z^t = \begin{bmatrix} x^t & \tilde{\Delta}^t \end{bmatrix}^\top$ (see Appendix A.1). Augmented dynamics, control, and measurement matrices $A_{\mathrm{aug}}, B_{\mathrm{aug}}, C_{\mathrm{aug}}$, and the noise vector $\eta_{\mathrm{aug}}$ are expressed in terms of $D$ and $E$. The feedforward and feedback energy costs are expressed as the LQG state and control costs respectively (Eq 12,

derived in Appendix A.2), where $Q = \begin{bmatrix} 0 & 0 \\ 0 & W_{\mathrm{f}} \end{bmatrix}$, and $R = W_{\mathrm{b}}$.

$$z^t = A_{\mathrm{aug}}\, z^{t-1} + B_{\mathrm{aug}}\, p^t + \eta_{\mathrm{aug}}^{t-1} \tag{10}$$

$$\tilde{\Delta}^t = C_{\mathrm{aug}}\, z^t \tag{11}$$

$$\min_p \lim_{T \to \infty} \frac{1}{T} \sum_{t=1}^{T} \left( \underbrace{z^{t\top} Q\, z^t}_{\text{state cost}} + \underbrace{p^{t\top} R\, p^t}_{\text{control cost}} \right) \tag{12}$$

Note that the inference cost is not explicitly added in the LQG objective function (Eq 12). However, using the *separation principle* [34], we show in Appendix A.3 that for a fixed $D$ and $E$ the LQG solution *automatically* minimizes the inference cost as well. The separation principle states that at each instant the observer needs to make an optimal estimate of the world state to form the optimal control. Furthermore, we show in Appendix A.4–A.6 that the LQG solution also provides the optimal $F, G, H, L$, and $\text{Cost}_{\text{tot}}$ in terms of $D$ and $E$. These are denoted as $F', G', H', L'$, and $\text{Cost}'_{\text{tot}}$ respectively. Finally, we numerically optimize $\text{Cost}'_{\text{tot}}$ with respect to $D$ and $E$ (as in Eq 9).

## 5    Results

In this paper we analyze our system for a one-dimensional world state to gain precise mathematical insight into the core computational problem. One useful perspective is that the feedback signal functions as a kind of self-control for the inference system. As we formulated the problem, this control has its own "action cost," and this allows us to use the known solutions for controllable systems to identify the optimal feedback. The control gain $L$ is therefore a fundamental parameter, as it indicates whether it is best to send a prediction ($L \neq 0$) or not ($L = 0$). The control gain always takes the opposite sign as the product of prediction-multiplier and observation-multiplier, ensuring that the prediction is subtracted from the observation, which thereby reduces the feedforward cost. However, when the optimal control gain is 0, the prediction-multiplier $D$ also becomes 0 so feedback noise does not corrupt the observation if no prediction is sent. Fig 2 shows how the optimal control gain, prediction-multiplier $D$ and observation-multiplier $E$ change with different parameters that define the constraints involved. We observe that there are cases where feedback and feedforward messages are useful, and others where messages are harmful.

### 5.1    Conditions
### for the Messages to be Useful

If the costs and noise are high, then it is not useful to send messages forward. To understand this quantitatively, we consider what happens without feedback ($D = 0$). In

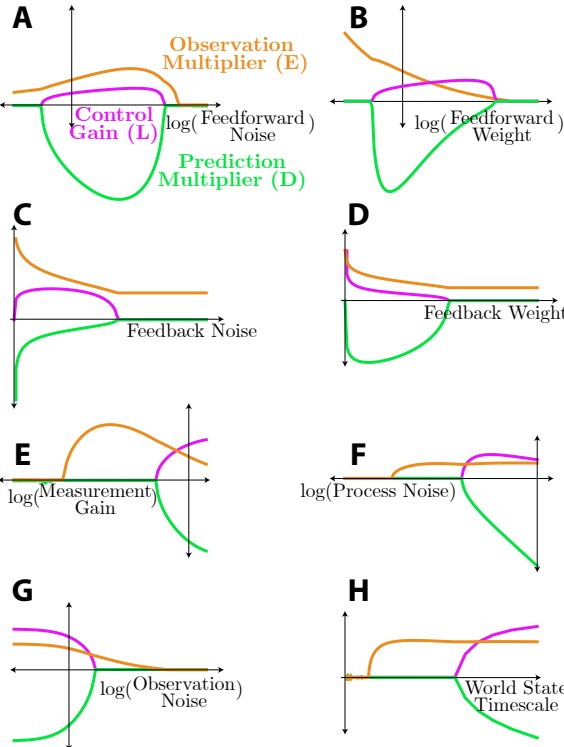

Figure 2: *The best use of predictions and observations depends on the world state dynamics and channels' noise costs. For tolerable feedback noise cost, as we increase either the A: feedforward noise, or the B: feedforward weight, the optimal strategy transitions from sending only feedforward, to predictive coding, back to only feedforward, and finally to no messages. For moderate feedforward noise cost, as we increase either the C: feedback noise, or the D: feedback weight, the optimal strategy transitions from predictive coding to feedforward messages only. When we increase either the E: measurement gain, or the F: process noise, or the H: world state timescale, the optimal strategy transitions from silence, to only feedforward messages, to predictive coding. G: Increasing the observation noise leads to the opposite sequence.*

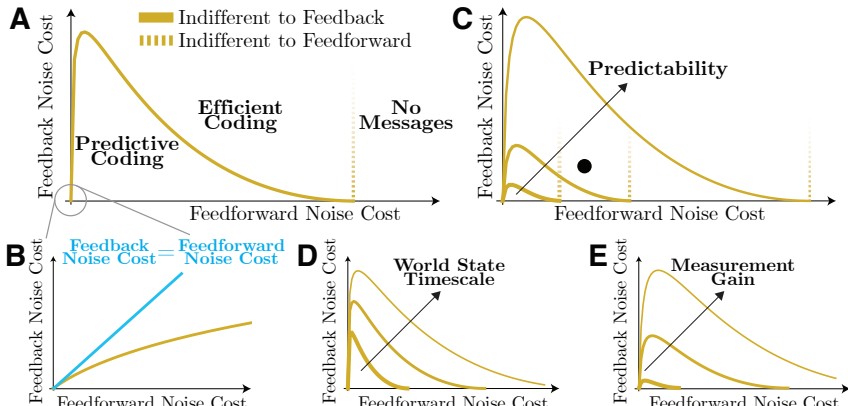

Figure 3: *Boundary curves divide the space of feedback and feedforward noise costs into regions of categorically different optimal strategies.* **A:** *Predictive coding is favored below the solid yellow line, feedforward efficient coding is favored above it, and no messages are favored to the right of the dashed line.* **B:** *The boundary curve determining whether feedback is useful lies below the line (cyan) along which the feedback and feedforward noise costs are equal, implying that for predictions to be useful the feedback channel must be cheaper and/or less noisy than the feedforward channel.* **C:** *With increasing predictability (thinner lines) of the world to be inferred, boundaries between coding strategies shift upwards and rightwards. As the predictability increases for a fixed value of channel parameters (dot), the optimal strategy transitions from sending no messages, to sending only feedforward messages, to sending and subtracting predictive feedback messages.* **D:** *An example of the shift in boundary curve with increasing predictability is shown for different world state timescales. As the timescale increases, the memory of the world state increases along with the SNR at the observation level, thereby increasing the predictability.* **E:** *A similar example is shown for different measurement gains. As the measurement gain increases, the observation SNR increases, thereby increasing the predictability.*

the absence of feedback, feedforward messages are worth its cost when the optimal observation multiplier $E$ at $D = 0$ is non-zero. We know that the derivative of the total cost at the optimal non-zero $E$, when it exists, should be 0. We set this derivative to 0 and then use Descartes' rule of signs to find when such a non-zero root exists. This yields the condition Eq 13. Sending even an infinitesimal feedforward message about the observation to the brain is worth its feedforward energy cost if and only if

$$(1 - A^2) \frac{U_{\mathrm{fn}}}{U_{\mathrm{s}}} < \frac{1}{1 + \frac{1}{\mathrm{SNR}_{\mathrm{o}}}} \tag{13}$$

where $U_{\mathrm{fn}} = W_{\mathrm{f}} \sigma_{\mathrm{f}}^2$ is the feedforward noise cost, i.e. the cost incurred by the noise variance in feedforward messages. $U_{\mathrm{s}} = \frac{\sigma_{\mathrm{p}}^2}{(1-A^2)}$ is the steady state signal power, and $\mathrm{SNR}_{\mathrm{o}} = \frac{\sigma_{\mathrm{p}}^2}{(1-A^2)} \frac{C^2}{\sigma_{\mathrm{o}}^2}$ is the observation Signal to Noise Ratio. When the left and right hand sides of Eq 13 are equal, the optimal strategy is indifferent between sending feedforward messages or not.

Next we consider the case where feedback is allowed. If the optimal prediction-multiplier is zero, then it is the same as no feedback, so by assumption the optimal observation-multiplier is non-zero if and only if feedforward is useful (Eq 13 holds). If the optimal prediction-multiplier is non-zero, then naturally the feedforward evidence must also be non-zero to have any signal worth predicting. Furthermore, when sending feedback is optimal, it must be worth sending some feedforward messages even if feedback is cut off. These purely feedforward messages might need to be smaller — as small as the residuals when we had feedback — but sending at least some information is evidently worth the cost, since it was worth paying that feedforward cost even when noisy feedback corrupted the inference. Thus, mathematically, feedforward messages are useful if and only if Eq 13 is true, both with and without feedback. Eq 13 implies that feedforward messages are worth their energy cost when either the observations are very reliable (high $\mathrm{SNR}_{\mathrm{o}}$), or when the relative cost of sending an arbitrarily small feedforward message is low (small $U_{\mathrm{fn}}/U_{\mathrm{s}}$).

Similarly, as shown in Appendix A.7, we find that feedback is valuable when

$$U_{\mathrm{bn}} < \Phi(U_{\mathrm{fn}}, A, \sigma_{\mathrm{p}}^2, \frac{C^2}{\sigma_{\mathrm{o}}^2}), \tag{14}$$

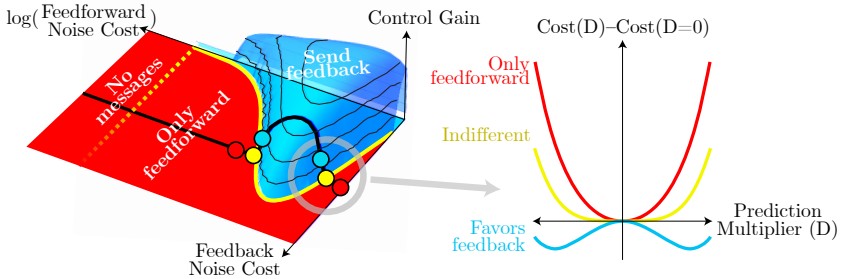

**Figure 4:** *Phase transition in the value of feedback.* Left*: Optimal control gain varies with the feedback and feedforward noise costs. As the feedforward noise cost increases (black line), the optimal strategy transitions non-monotonically from sending only feedforward messages, to predictive coding with suppressive feedback, back to sending only feedforward messages, and eventually to sending no messages. The yellow curve on the panel is the phase transition for feedback: the locus of points where the optimal strategy is indifferent to sending feedback (green area) or not (red area). Near the critical point (colored circles), the loss function goes through a phase transition [35], seen at* Right*: The cost function has a minimum at Prediction-multiplier $D = 0$ for low feedforward noise cost, favoring no feedback (red). But as the feedforward noise cost increases from low to moderate values, this extremum switches to a maximum, with nearby minima that favor feedback with nonzero $D$ (green). Right at the transition point, the system becomes indifferent to feedback (yellow). Another phase transition occurs in the opposite direction as the feedforward noise cost increases further from moderate to high.*

where $\Phi$ is a complicated function that involves solving for a quartic equation. Below in section 5.2 we will interpret the behavior of this function. $U_{\mathrm{bn}} = W_{\mathrm{b}}\sigma_{\mathrm{b}}^2$ is the feed**b**ack **n**oise cost, i.e. the cost incurred by the noise variance in feedback messages. In Fig 3A, $\Phi$ (solid yellow line) is plotted as a function of the feedforward noise cost. Where Eq 14 holds as an equality (feedback and feedforward noise costs that correspond to points on the yellow line), an optimal observer is indifferent about sending feed*back* messages. Sending feedback is useful below that line, and harmful above it.

As stated above, when sending feedback is optimal (Eq 14 satisfied), it must be worth sending feedforward messages even if feedback is cut off (Eq 13 holds true). This is indeed observed in Fig 3A, where the dashed line is the locus of points where the optimal strategy is indifferent to sending feed*forward* messages: to the left, sending feedforward messages is useful, and to the right, it is not. As expected, the region where feedback is useful is also where feedforward messages are useful. Also, for very small feedback noise cost, the only case where sending feedback becomes harmful is when sending even an infinitesimal feedforward message is not useful either. As a result, the feedback indifference curve ends exactly at the precise feedforward noise cost where the optimal strategy is indifferent to sending feedforward messages. Fig 3B shows a magnified version where feedforward and feedback products are near zero. At the origin, where the products equal zero, feedforward and feedback messages are free and/or noiseless. A noiseless message can be arbitrarily small and still convey information, so the cost can be arbitrarily small. In this case, there is nothing to gain or lose by subtracting predictions, so even free feedback makes no difference to costs or inferences. Thus the feedback indifference curve starts at the origin. Fig 3B also shows that for feedback to be useful, the feedback channel should be cheaper and/or less noisy than the feedforward channel. Note that this is true regardless of sensory parameter values (world state timescale, measurement gain, etc.).

The exact optimal multipliers for the optimization in Eq 9 depends on all system parameters (Eqs 1–7): feedforward and feedback noise variances and weights, the measurement gain, world state timescale, the process and observation noise variances. However, from Eqs 13–14, we see that the optimal categorical *strategy* — which multipliers are nonzero — depends only on certain combinations of these parameters. For example, the phase transition depends on the measurement gain and observation noise only through their ratio, ($\mathrm{SNR_o}$), which determines the quality of observations. Specifically, in Appendix A.8–A.9, we prove that the feedforward (feedback) noise variance and feedforward (feedback) weights affect the phase transition only through their product, the feedforward (feedback) noise cost. The core reason is the following: If any (possibly suboptimal) multiplier is nonzero for one value of noise and cost weight, then there is a whole *family* of nonzero multipliers for noises and cost weights that are scaled inversely, keeping their product (the noise cost) fixed — where every member of this family has *identical* costs and signals. Thus if we optimize the multipliers for one noise and weight, and find it is nonzero, then the optimal multipliers will still be nonzero for this

entire family of inversely scaled noises and weights. In other words, whether feedforward or feedback is useful depends on the noise costs.

## 5.2 Transitions in the Optimal Strategy

Having seen above how there are categorically different optimal strategies for computationally constrained inference, we now examine how individual parameters move the system between these strategies. We broadly group parameters into three categories: feedforward, feedback, and sensory.

**Feedforward parameters:** We first consider the case of low to moderate feedback noise cost. Fig 4 shows the transition between optimal strategies as a function of the feedforward and feedback noise costs. The black line on Fig 4 left traces the transition as we increase the feedforward noise cost for a fixed feedback noise cost. For low values of the feedforward noise cost, feedforward messages are almost free, so the system does not save appreciable resources by sending predictions; even worse, noisy feedback would corrupt the signal with noise. Thus for low feedforward noise costs it is optimal to send no predictions. As this cost increases, at some point it becomes equally valuable to send or withhold a feedback message. For higher feedforward noise costs, we cross the point of indifference, to where feedforward messages are important yet their channel is not economical by itself. Predictive feedback then becomes preferable, even when accounting for additional feedback noise.

As feedforward noise cost increases, reliable transmission through the channel becomes less affordable. As a consequence, the inference degrades. Upon increasing the feedforward noise cost beyond a certain point, the inference becomes so poor that it is no longer possible to make a good prediction worthy of the cost it incurs. Thus the system resorts to sending only feedforward messages. For similar reasons, as the world becomes more predictable (Fig 3C), there is a wider range of feedforward and feedback noise costs for which sending predictions is optimal. The predictability can be increased either by lengthening the world state timescale, or by enhancing the observation SNR, enabling better inferences and thereby better predictions. Figs 3D–E demonstrates this for different world state timescales and measurement gains respectively. However, for large enough feedforward noise cost, it is best to remain silent as sending even feedforward messages alone becomes too expensive/noisy.

Fig 2A and B show how the optimal control gain $L$, prediction-multiplier $D$ and observation-multiplier $E$ change with feedforward noise and feedforward weight respectively. As only the product of noise variance and weight determines the transition in optimal strategy, we observe the same trend as either one of them increases. However, these parameters actually exhibit different effects away from the points of transitions. For example, when feedforward noise is close to $0$, the system can save costs by attenuating the signal ($E \to 0$) and still beat the negligible feedforward noise. In contrast, in the extreme case that the feedforward messages are nearly free ($W_\mathrm{f} \to 0$), the observation-multiplier increases arbitrarily ($E \to \infty$) to dominate any feedforward noise, improving the inference at no cost. These limiting cases provide helpful intuitions. Naturally, the challenging regime is in the middle, where there are real tradeoffs to make, and where feedback becomes relevant.

For the limiting case where feedback noise cost is extremely high, the feedback channel is too expensive/noisy to be used. Hence, as we slowly increase the feedforward noise cost from zero, we start by sending just the feedforward messages until the feedforward channel is too expensive/noisy, at which point we no longer send any messages.

**Feedback parameters:** Analogous to the analysis described for feedforward parameters, here we discuss how the optimal strategy changes as we increase the feedback noise cost, for different levels of feedfoward noise cost. We first consider the interesting case of low to moderate feedfoward noise cost. For low values of feedback noise cost we are in the regime where feedback is cheap and/or noiseless, and hence it is beneficial to send both feedback and feedforward messages. But when the feedback noise cost becomes too high, sending predictions become too expensive/noisy, so it is best send only feedforward messages. This is shown in Fig 2C, D. Analogous to the feedforward case, although the same transition in optimal strategy is observed as we increase either the feedback noise or weight, these parameters actually exhibit different effects away from that transition. For example, $L \to 0$ when feedback noise is close to $0$, and $L \to -\infty$ when feedback weight is close to $0$. Finally, in the case of extremely high feedfoward noise cost, it is not worth sending even an infinitesimal feedforward message, so the optimal strategy is to not send any messages.

**Sensory parameters:** Fig 3C shows the boundary between strategies for three systems with different levels of predictability. For a fixed value of feedback and feedforward noise costs (dot), the optimal

strategy changes with the predictability. With unpredictable dynamics (thick curve), the dot lies in the region where sending no messages is optimal, since the feedforward channel is relatively poor (right of the thick dashed line). For a slightly higher predictability, the dot lies within the region where feedforward-only messages are optimal (above medium curve, but left of the corresponding dashed line). And for even higher predictability, the dot lies in the region where predictive coding is optimal (below the thin curve). Hence, as the predictability increases, there is a transition from sending no messages, to sending just feedforward, to sending both feedback and feedforward messages.

Similarly, if either the measurement gain or the process noise increases, it would yield a higher observation SNR, which would then improve the inference and thereby predictability. Figs 2E–F show the resultant transitions from no messages, to feedforward-only messages, then to predictive coding, with increasing measurement gain or process noise. Increasing observation noise has the opposite effect, making observations less reliable and thereby reducing the predictability. This is shown in Fig 2G. Finally, increasing the world state timescale increases the predictability since it increases both the memory of the system, and the observation SNR. Fig 2H reveals the familiar sequence of transitions in strategy, supporting our core intuitions about when feedback is valuable.

### 5.3 Anatomical and functional differences can lead to different channel noise costs

We characterized the properties of feedforward/feedback channel in terms of the channel noise and weight of energetic cost, and showed that the phase transition depends on the feedforward/feedback channel noise costs. But why would channels have different noise costs? Biologically, asymmetries in noise costs can arise from anatomical or functional constraints. For example, Dale's law [36] states that each neuron makes only excitatory or inhibitory synapses onto its targets. However, long-range feedback connections in the brain are excitatory [37, 38]. This implies that to subtract predictions using long-range inhibitory feedback, the signals must pass through two neurons: long-range excitation that is inverted by short-range inhibition. In contrast, feedforward signals can proceed directly to their targets. The sign inversion neurons induce additional noise in the feedback pathway. Below we review biological evidence that feedback and feedforward channels have distinct anatomical and functional properties.

To see how channels with different anatomical structures can yield different channel noise costs, contrast two versions of a channel with equal numbers of neurons, all having identical energetic weights and noise variances. One version has a single population of neurons that linearly encodes the message to be sent; the channel output simply decodes this population's representation. We call this channel the non-sequential case (Appendix Fig 9A). The second version is the sequential case (Appendix Fig 9B), and uses two populations sequentially connected in series, one after the other, like long-range inhibition comes from long-range excitation followed by short-range inhibition. The first population linearly encodes the message to be sent, which is then decoded and then encoded by the second population; the channel's output is a decoding of the second population.

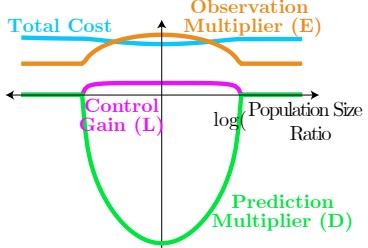

Figure 5: *Allocation of neurons between two sequentially connected populations of the feedback channel affects when sending feedback is useful.*

Appendix A.10.1 shows that for the non-sequential case with one population, as we scale up the number of neurons, the channel's weight scales up and the noise scales down inversely. Their product, the channel noise cost, does not change with the number of neurons, yielding the same optimal categorical strategy for a given noise and weight per neuron. For the sequential case with two populations, we show in Appendix A.10.2 that the channel noise cost again does not depend on the total number of neurons, but does depend on the ratio of population sizes. Furthermore, the total cost changes non-monotonically with population size ratio for the sequential case, and increases monotonically with total number of neurons for both sequential and non-sequential cases. Fig 5 shows the optimal strategy changing with the population size ratio of the feedback channel, for a fixed total number of neurons. We see that the feedback is not useful when the ratio is strongly skewed (away from 1).

From the above example, we saw that channels with different anatomical structures can yield different noise costs, and thereby different optimal strategies. Biologically, there exist differences between the

anatomy or functional properties of the feedback and feedforward channels. Anatomically, feedback projections generally outnumber feedforward ones by 2:1, but the feedforward pathways compensate with greater weight, dominating at short cortical distances and leveling off at longer distances [39]. Functionally, the sparser, stronger feedforward channels therefore have less ability to average away the noise, leading to a relatively higher amount of metabolically expensive activity caused by noise, and thus to a higher noise cost. This could establish conditions under which feedback is useful.

## 6   Discussion

In this paper, we define a new class of dynamic optimization tasks that reflect essential biological constraints on inference in the brain, by including cost and noise for each recurrently connected computational element. We solve this optimization problem by modeling inference as a control problem with prediction as self-control. The resultant optimization provides nontrivial predictions for when we expect suppressive feedback as a function of biological constraints, computational costs, and world dynamics.

Predictive coding is a promising theory for brain computations [40–44], and a variety of experimental studies have indeed shown predictive suppression effects in neural responses [45–52]. However, neural responses vary in how much they are suppressed depending on the context and SNR. Variants aiming to explain these effects all neglect the computational costs associated with predictive coding, even though those costs were part of the motivation for saving metabolic costs in the first place.

Past work described temporal decorrelation as optimal encoding strategy, subtracting a slow, predictable component from the recent measured signal, implementing a form of predictive coding [15, 53]. Like our results, they too observed that this strategy changes with the SNR. Efficient coding shows that mean responses are fully subtracted at all SNRs, but with different temporal kernels [15, 53]. Later studies of predictive coding observed that for low SNR, temporal decorrelation is no longer the best strategy [18]. Our work generalizes these results to account not only for feedforward noise and costs in sensory coding, but also for noise and costs in recurrent computation. In particular, the nonmonotonic dependence of optimal feedback on feedforward noise costs is a novel consequence of our framework with no analog in previous studies. We could also imagine generalizing our results to adaptive strategies, like [31], where optimal adaptation could implement our predicted changes in the optimal suppression as sensory statistics change. Conversely, by considering the computational costs of adaptation itself (and not just the activity resulting from adapted computation), we may find conditions where adapting is not beneficial.

Although we contend that it is costly to send predictions, if these predictions are based on inferences that are already being computed for a task, then is it really an extra cost to send them as predictions? Yes, because distinct neurons are used to send feedforward and feedback signals [54], so the brain might pay costs twice for the same inference. When beneficial, the brain could avoid the duplicate feedback costs by silencing the feedback neurons, and send its inferences to higher brain areas only through its feedforward neurons. A related question arises for feedback suppression, which is presumably implemented through inhibitory neurons. What is the benefit of turning on an inhibitory neuron just in order to turn off an excitatory neuron? Excitatory neurons outnumber inhibitory neurons by 4:1 [55], so few inhibitory neurons can suppress many excitatory ones, amounting to a potentially substantial savings. These biological constraints could be used in future elaborations of computationally constrained inference models.

Our mathematical system's crucial theoretical parameters relate to biological quantities in three ways: as context-dependent parameters (time constant of the stimulus dynamics, sensory SNR), as neural parameters that could potentially be manipulated experimentally (feedforward and feedback SNR), and as developmentally-fixed parameters of the system (feedforward and feedback architecture). Testing our predicted dependence on these three types of parameters requires different considerations. First, the context-dependent parameters are by far the easiest to manipulate experimentally. One can control the stimulus to adjust the observations' SNR and the world state timescale, and measure whether any response suppression is

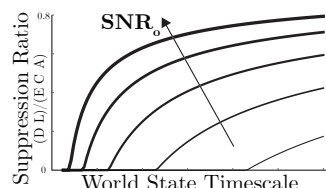

Figure 6: *World timescale and observation SNR affect predictive response suppression. SNR increases with line thickness.*

modulated by these controlled parameters as shown in Fig 6. Second, the internal computational parameters like neural noise may be controllable through stimulus changes or experimental techniques

of causal manipulation. For example, electrical or optogenetic stimulation could directly inject neural variability. Another approach could be based on the fact that neural noise variance tends to increase with firing rate, so one could test our predictions about computational noise by elevating firing rates. Such methods could include providing a background sensory stimulus, or direct neural stimulation to increase baseline activity. In any case, it would be important to apply either natural manipulations or chronic unnatural ones to give the brain enough time and experience to optimize its computations. And finally, for the developmentally fixed parameters like architecture, we cannot easily alter the system experimentally, and it may be unreasonable to assume that properties in a real biological system can be unambiguously mapped to any particular value in our abstract theoretical system. However, we can compare *between* brain systems or species with architectural difference, and test whether functional properties covary with those differences as predicted by our theory. Thus it may be fruitful to compare between brain areas with different architectures, such as 3-layer paleocortex (e.g. olfactory cortex) versus 6-layer neocortex (e.g. auditory or visual cortex), or within neocortical areas with different properties. It may also be fruitful to compare between organisms with different architectures (e.g. mammals versus reptiles). These architectures have different microcircuits with distinct feedforward and feedback projection neurons. For example, feedback in different visual and motor cortical systems tends to target different inhibitory cells [56] which have different noise levels [57, 58]. We predict that systems with more feedback noise (e.g. SOM versus VIP) would have less predictive suppression, as observed experimentally [56].

Our theory of *inference as control* could also be used in designing energy-efficient systems in other domains where the estimation takes place under constraints. For example, low-power applications of inference and control, such as drones or remote sensing, often place constraints on dynamic range and noise (e.g. quantization noise) for all signals, and could thus benefit from optimizations based on our model.

**Limitations.** There are several interesting avenues for generalizing our theory. The proposed *inference as control* method can be extended to the more general case of observer taking external actions in addition to the internal predictions, in order to maximize external rewards while minimizing both external action costs (e.g. movements) and internal computational costs. This could be modeled as a typical LQG control problem for external variables by including appropriate entries in matrices $Q$, $R$, and $B_{\mathrm{aug}}$ in addition to the internal controls we use for predictive coding. Though our results are limited to one-dimensional linear dynamics with uncorrelated noise, our future work will explore these computational constraints in more complex multivariate and nonlinear tasks. Accounting for graph-structured inferences [59] and controllability [60] may provide additional constraints on the brain's inference processes, and additional predictions for experiments.

Residuals between predictions and observations are useful not just for improving inferences, but also for learning, which this paper does not address. In principle, a multivariate generalization could explain such computations as hierarchical inferences or adaptation where slower changes are also subject to prediction. However, unlike adaptation, learning occurs out of equilibrium with the environment, and accounting for transient responses under nonstationary statistics will require an extension to our theory. Overall, this work points a way towards expanding theories of predictive coding and efficient coding to unify inference, learning, and control in biological systems.

**Acknowledgments.** The authors thank Itzel Olivos Castillo, Ann Hermundstad, Krešimir Josić, and Paul Schrater for useful discussions. This work was supported in part by NSF CAREER grant 1552868, the McNair Foundation, and AFOSR grant FA9550-21-1-0422 in the Cognitive and Computational Neuroscience program.

**Competing interests.** XP is a founder of Upload AI, LLC, a company in which he has related financial interests.

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
