# A   Appendix

## A.1   Mapping the inference tracking dynamics onto an LQG control problem

Substituting Eqs 1- 2, and 4-5 in Eq 6.

$$
\begin{aligned}
\tilde{\Delta}^t &= \Delta^t + \eta_{\mathrm{f}}^t \\
&= (D\,\tilde{p}^t + E\,o^t) + \eta_{\mathrm{f}}^t \\
&= D\,\tilde{p}^t + E\,(C\,x^t + \eta_{\mathrm{o}}^t) + \eta_{\mathrm{f}}^t \\
&= D\,\tilde{p}^t + E\,C\,x^t + E\,\eta_{\mathrm{o}}^t + \eta_{\mathrm{f}}^t \\
&= D\,\tilde{p}^t + E\,C\,(A\,x^{t-1} + \eta_{\mathrm{p}}^t) + E\,\eta_{\mathrm{o}}^t + \eta_{\mathrm{f}}^t \\
&= D\,\tilde{p}^t + E\,C\,A\,x^{t-1} + E\,C\,\eta_{\mathrm{p}}^t + E\,\eta_{\mathrm{o}}^t + \eta_{\mathrm{f}}^t \\
&= D\,(p^t + \eta_{\mathrm{b}}^t) + E\,C\,A\,x^{t-1} + E\,C\,\eta_{\mathrm{p}}^t + E\,\eta_{\mathrm{o}}^t + \eta_{\mathrm{f}}^t \\
&= E\,C\,A\,x^{t-1} + D\,p^t + E\,C\,\eta_{\mathrm{p}}^t + E\,\eta_{\mathrm{o}}^t + D\,\eta_{\mathrm{b}}^t + \eta_{\mathrm{f}}^t
\end{aligned}
\tag{15}
$$

Combining Eqs 1 and 15, we write the dynamics of augmented state $z^t = \begin{bmatrix} x^t & \tilde{\Delta}^t \end{bmatrix}^\top$ as in Eq 16. The noisy residual can now be written as a partial observation of the augmented state (Eq 17).

$$
z^t = \begin{bmatrix} x^t \\ \tilde{\Delta}^t \end{bmatrix} = \overbrace{\begin{bmatrix} A & 0 \\ E\,C\,A & 0 \end{bmatrix}}^{A_{\mathrm{aug}}} \underbrace{\begin{bmatrix} x^{t-1} \\ \tilde{\Delta}^{t-1} \end{bmatrix}}_{z^{t-1}} + \overbrace{\begin{bmatrix} 0 \\ D \end{bmatrix}}^{B_{\mathrm{aug}}} p^t + \overbrace{\begin{bmatrix} I & 0 & 0 & 0 \\ E\,C & E & D & I \end{bmatrix} \begin{bmatrix} \eta_{\mathrm{p}}^t \\ \eta_{\mathrm{o}}^t \\ \eta_{\mathrm{b}}^t \\ \eta_{\mathrm{f}}^t \end{bmatrix}}^{\eta_{\mathrm{aug}}^{t-1}}
\tag{16}
$$

$$
\tilde{\Delta}^t = \underbrace{\begin{bmatrix} 0 & I \end{bmatrix}}_{C_{\mathrm{aug}}} \underbrace{\begin{bmatrix} x^t \\ \tilde{\Delta}^t \end{bmatrix}}_{z^t}
\tag{17}
$$

Rewriting Eqs 16-17 using concise notation, we get the LQG dynamics equations where prediction $p$ is treated as the control.

$$
\begin{aligned}
z^t &= A_{\mathrm{aug}}\,z^{t-1} + B_{\mathrm{aug}}\,p^t + \eta_{\mathrm{aug}}^{t-1} \\
\tilde{\Delta}^t &= C_{\mathrm{aug}}\,z^t
\end{aligned}
$$

## A.2   Feedforward and feedback energy costs expressed as the LQG state and control costs respectively

We express the feedforward and feedback energy costs as the LQG state and control costs respectively. For the derivation, we introduce the matrices $Q = \begin{bmatrix} 0 & 0 \\ 0 & W_{\mathrm{f}} \end{bmatrix}$, and $R = W_{\mathrm{b}}$. We take the feedback noise $\eta_{\mathrm{b}}$ as i.i.d $\sim \mathcal{N}(0, \Sigma_{\mathrm{b}})$.

$$
\begin{aligned}
\min_p \lim_{T\to\infty} \frac{1}{T} \sum_{t=1}^{T} \left( \tilde{\Delta}^{t\top} W_{\mathrm{f}}\, \tilde{\Delta}^t + \tilde{p}^{t\top} W_{\mathrm{b}}\, \tilde{p}^t \right) &= \min_p \lim_{T\to\infty} \frac{1}{T} \sum_{t=1}^{T} \left( \tilde{\Delta}^{t\top} W_{\mathrm{f}}\, \tilde{\Delta}^t + (p^t + \eta_{\mathrm{b}}^t)^\top W_{\mathrm{b}}\, (p^t + \eta_{\mathrm{b}}^t) \right) \\
&= \min_p \lim_{T\to\infty} \frac{1}{T} \sum_{t=1}^{T} \left( \tilde{\Delta}^{t\top} W_{\mathrm{f}}\, \tilde{\Delta}^t + p^{t\top} W_{\mathrm{b}}\, p^t + \eta_{\mathrm{b}}^{t\top} W_{\mathrm{b}}\, \eta_{\mathrm{b}}^t \right) \\
&= \min_p \lim_{T\to\infty} \frac{1}{T} \sum_{t=1}^{T} \left( \tilde{\Delta}^{t\top} W_{\mathrm{f}}\, \tilde{\Delta}^t + p^{t\top} W_{\mathrm{b}}\, p^t \right) + Tr(W_{\mathrm{b}}\,\Sigma_{\mathrm{b}}) \\
&= \min_p \lim_{T\to\infty} \frac{1}{T} \sum_{t=1}^{T} \left( z^{t\top} Q\, z^t + p^{t\top} R\, p^t \right) + Tr(W_{\mathrm{b}}\,\Sigma_{\mathrm{b}}) \\
&= \min_p \lim_{T\to\infty} \frac{1}{T} \sum_{t=1}^{T} \left( \underbrace{z^{t\top} Q\, z^t}_{\text{state cost}} + \underbrace{p^{t\top} R\, p^t}_{\text{control cost}} \right)
\end{aligned}
$$

## A.3 Minimizing the total energy cost also minimizes the inference cost, for a fixed D & E

The LQG problem of interest is described by Eqs 10–12. The separation principle states that the solution for LQG problem includes an estimation part and a control part, both of which jointly form the optimal solution even if treated separately. The estimation part of LQG solution minimizes the average squared norm error between estimated state $\hat{z}$ and the true state $z$, as shown in the left side of equality in Eq 18. Note that at any time $t$, we get to observe all the evidence up until that time (i.e $\tilde{\Delta}^1, \tilde{\Delta}^2,.., \tilde{\Delta}^t$). This enables the simplification from Eq 18 to 19, because the optimal estimate $\hat{\tilde{\Delta}}^t$ given $\tilde{\Delta}^t$ is trivially $\tilde{\Delta}^t$ itself. Hence, from Eq 20, we conclude that although we define the LQG objective function to be just the total energy cost, the LQG solution that minimizes the total energy cost also minimizes the inference cost for a fixed $D$ and $E$.

$$\lim_{T\to\infty} \frac{1}{T} \sum_{t=1}^{T} (z^t - \hat{z}^t)^\top (z^t - \hat{z}^t) = \lim_{T\to\infty} \frac{1}{T} \sum_{t=1}^{T} (\begin{bmatrix} x^t \\ \tilde{\Delta}^t \end{bmatrix} - \begin{bmatrix} \hat{x}^t \\ \hat{\tilde{\Delta}}^t \end{bmatrix})^\top (\begin{bmatrix} x^t \\ \tilde{\Delta}^t \end{bmatrix} - \begin{bmatrix} \hat{x}^t \\ \hat{\tilde{\Delta}}^t \end{bmatrix}) \tag{18}$$

$$= \lim_{T\to\infty} \frac{1}{T} \sum_{t=1}^{T} (x^t - \hat{x}^t)^\top (x^t - \hat{x}^t) \tag{19}$$

$$= \text{Cost}_{\text{inf}} \tag{20}$$

## A.4 LQG estimation solution to find $F$, $G$, and $H$ in terms of $D$ and $E$

The optimal estimate of the augmented state based on its noisy partial observations is given as

$$\hat{z}^t = (I - K\, C_{\text{aug}}) A_{\text{aug}}\, \hat{z}^{t-1} + K\, \tilde{\Delta}^t + (I - K\, C_{\text{aug}}) B_{\text{aug}}\, p^t \tag{21}$$
$$K = \check{\Sigma}\, C_{\text{aug}}^T (C_{\text{aug}}\, \check{\Sigma}\, C_{\text{aug}}^\top)^{-1},$$

where $\check{\Sigma}$ is the solution to a discrete algebraic Riccati equation (Eq 22), with $W$ as the covariance of noise vector $\eta_{\text{aug}}$.

$$\check{\Sigma} = A_{\text{aug}}\, \check{\Sigma}\, A_{\text{aug}}^\top + W - A_{\text{aug}}\, \check{\Sigma}\, C_{\text{aug}}^\top (C_{\text{aug}}\, \check{\Sigma}\, C_{\text{aug}}^\top)^{-1} C_{\text{aug}}\, \check{\Sigma}\, A_{\text{aug}}^\top \tag{22}$$

For brevity, we rewrite Eq 21 in terms of block matrics using the following substitutions,

$$F_{\text{aug}} = (I - K\, C_{\text{aug}}) A_{\text{aug}},\ H_{\text{aug}} = (I - K\, C_{\text{aug}}) B_{\text{aug}}$$
$$\begin{bmatrix} \hat{x}^t \\ \tilde{\Delta}^t \end{bmatrix} = \begin{bmatrix} (F_{\text{aug}})_{11} & (F_{\text{aug}})_{12} \\ (F_{\text{aug}})_{21} & (F_{\text{aug}})_{22} \end{bmatrix} \begin{bmatrix} \hat{x}^{t-1} \\ \tilde{\Delta}^{t-1} \end{bmatrix} + \begin{bmatrix} (K)_1 \\ (K)_2 \end{bmatrix} \tilde{\Delta}^t + \begin{bmatrix} (H_{\text{aug}})_1 \\ (H_{\text{aug}})_2 \end{bmatrix} p^t \tag{23}$$

yielding

$$\hat{x}^t = (F_{\text{aug}})_{11}\, \hat{x}^{t-1} + (F_{\text{aug}})_{12}\, \tilde{\Delta}^{t-1} + (K)_1\, \tilde{\Delta}^t + (H_{\text{aug}})_1\, p^t \tag{24}$$
$$\tilde{\Delta}^t = (F_{\text{aug}})_{21}\, \hat{x}^{t-1} + (F_{\text{aug}})_{22}\, \tilde{\Delta}^{t-1} + (K)_2\, \tilde{\Delta}^t + (H_{\text{aug}})_2\, p^t.$$

Since the states are Markovian, consequently the optimal estimates are Markovian as well. This implies that $\hat{x}^t$ given $\hat{x}^{t-1}$, does not depend on $\tilde{\Delta}^{t-1}$. Likewise, $\tilde{\Delta}^t$ given $\tilde{\Delta}^t$, does not depend on $\hat{x}^{t-1}, \tilde{\Delta}^{t-1}$, and $p^t$. The above two arguments can be used to deduce that

$$(F_{\text{aug}})_{12} = 0,\ (F_{\text{aug}})_{21} = 0,\ (F_{\text{aug}})_{22} = 0\ (K)_2 = I,\ (H_{\text{aug}})_2 = 0.$$

which we also verified empirically for the one dimensional case. Substituting $(F_{\text{aug}})_{12} = 0$ in Eq 24 and comparing it with Eq 7, we get the optimal solutions for $F$, $G$, and $H$ in terms of $D$ and $E$, denoted as $F'$, $G'$, and $H'$ respectively.

$$F' = (F_{\text{aug}})_{11},\ G' = (K)_1,\ H' = (H_{\text{aug}})_1$$

## A.5 LQG control solution to find $L$ in terms of $D$ and $E$

The optimal LQG control is given as

$$p^t = L_{\text{aug}}\, \hat{z}^{t-1} \tag{25}$$
$$L_{\text{aug}} = -(R + B_{\text{aug}}^\top P\, B_{\text{aug}})^{-1} B_{\text{aug}}^\top P\, A_{\text{aug}} \tag{26}$$

where $P$ is the solution of a discrete algebraic Riccati equation (Eq 27).

$$P = Q + A_{\text{aug}}^\top P A_{\text{aug}} - A_{\text{aug}}^\top P B_{\text{aug}}(R + B_{\text{aug}}^\top P B_{\text{aug}})^{-1} B_{\text{aug}}^\top P A_{\text{aug}} \qquad (27)$$

As shown in Eq 16, the last set of columns in $A_{\text{aug}}$ are all zero vectors. As a consequence, since $A_{\text{aug}}$ is the last multiplier in the right side of equality in Eq 25, the last set of columns of $L_{\text{aug}}$ would also be zero vectors. More specifically, Eq 25 can be further simplified by writing $L_{\text{aug}}$ in terms of block matrices in the following way

$$p^t = [(L_{\text{aug}})_1 \quad (L_{\text{aug}})_2]\begin{bmatrix}\hat{x}^{t-1} \\ \tilde{\Delta}^{t-1}\end{bmatrix} = (L_{\text{aug}})_1 \ \hat{x}^{t-1} + (L_{\text{aug}})_2 \ \tilde{\Delta}^{t-1}, \text{ then } (L_{\text{aug}})_2 = 0$$

$$\implies p^t = (L_{\text{aug}})_1 \ \hat{x}^{t-1}$$

Hence, the optimal prediction is linearly related to the estimate through $(L_{\text{aug}})_1$. For consistency in notations, we denote $(L_{\text{aug}})_1$ as $L'$, yielding

$$p^t = L' \ \hat{x}^{t-1}, \text{ where } L' = (L_{\text{aug}})_1.$$

### A.6 Closed form expressions for total cost in terms of $D$ and $E$

The optimal inference cost for fixed $D$ and $E$, which is the same as the augmented state's estimation error, is $\text{Cost}'_{\text{inf}} = Tr(\Sigma)$. Where $\Sigma$ is the augmented state's estimation error covariance matrix, such that

$$\Sigma = \check{\Sigma} - \check{\Sigma} \ C_{\text{aug}}^\top (C_{\text{aug}} \ \check{\Sigma} \ C_{\text{aug}}^\top)^{-1} \ C_{\text{aug}} \ \check{\Sigma}. \qquad (28)$$

The optimal total energy cost for a given $D$ and $E$, which is the optimal LQG cost, is given as $\text{Cost}'_{\text{energy}} = Tr(Q \ \Sigma) + Tr(P(\check{\Sigma} - \Sigma))$. As $\Sigma$ is the error covariance matrix in estimating $z^t$, and since the noisy residual $\tilde{\Delta}^t$ is fully observable at any time point, the corresponding lower block matrix in $\Sigma$ will be zero. Also, by definition in Eq 12, we know that the upper block matrix in $Q$ is zero. As a consequence, $Tr(Q \ \Sigma)$ would always be zero. Resulting in $\text{Cost}'_{\text{energy}} = Tr(P(\check{\Sigma} - \Sigma))$. Combining the inference cost and energy cost, we have the total cost in terms of $D$ and $E$ as

$$\text{Cost}'_{\text{tot}} = Tr(\Sigma) + Tr(P(\check{\Sigma} - \Sigma)) \qquad (29)$$

### A.7 Condition for useful feedback

It can be algebraically shown that the total cost is an even function in both $D$ and $E$ and hence can be written as a function of $\check{D} = D^2$ and $\check{E} = E^2$. This limits the function domain to non-negative values, thereby reducing the search space for numerical optimization. We empirically observe that the cost function now has just one minimum when plotted with $\check{D}$ and $\check{E}$, and that the negative of gradient at any point always directs toward this minimum.

Let $\check{E}'$ denote the optimal $\check{E}$ that minimizes the total cost at $\check{D} = 0$. $\check{E}'$ can be computed by equating the derivative of cost with respect to $\check{E}$ to 0, at $\check{D} = 0$ (Eq 30).

$$\left.\frac{\partial \text{Cost}'_{\text{tot}}}{\partial \check{E}}\right|_{\check{D}=0, \check{E}=\check{E}'} = 0 \qquad (30)$$

We define $\psi$ as the derivative of total cost with respect to $\check{D}$, evaluated at $\check{D} = 0$ and $\check{E} = \check{E}'$ (Eq 31).

$$\psi = \left.\frac{\partial \text{Cost}'_{\text{tot}}}{\partial \check{D}}\right|_{\check{D}=0, \check{E}=\check{E}'} \qquad (31)$$

Since the negative of gradient always points to the minimum, the following holds true: If $\psi$ is negative, then the negative of gradient points towards non-zero $\check{D}$, implying the optimal $\check{D}$ is non-zero. However, given that the domain of $\check{D}$ is non-negative, if $\psi$ is non-negative then the optimal $\check{D}$ is 0. Therefore, the optimal $\check{D}$ is non-zero if and only if $\psi < 0$. These conditions can be concisely written as

$$U_{\text{bn}} < \Phi(U_{\text{fn}}, A, \sigma_{\text{p}}^2, \frac{C^2}{\sigma_{\text{o}}^2}). \qquad (32)$$

where $\Phi$ is a complicated function involving the roots $\check{E}'$ of the quartic equation Eq 30 substituted into Eq 31. Ultimately we use these analytic expressions to solve numerically for the remaining conditions on the optimal prediction-multiplier $D$ and observation-multiplier $E$.

## A.8 Channel communication in terms of pre and post multipliers

Let $s$ be a random variable describing the signal of interest, following the distribution $\mathcal{N}\left(0, \sigma_{\text{signal}}^2\right)$. The signal is pre-multiplied with $m_1$ before passing it through an additive Gaussian noise channel. The channel noise is described using the random variable $\eta \sim \mathcal{N}\left(0, \sigma_{\text{noise}}^2\right)$. The message sent over the channel would then be $m_1\, s + \eta$. The message is then post-multiplied with $m_2$ to form the channel output, $\boldsymbol{\pi}(s, m_1, m_2, \sigma_{\text{noise}}^2) = m_2\left(m_1\, s + \eta\right)$. Let $w$ denote the weight of channel's energetic cost. Then the energetic cost per message is given as,

$$\boldsymbol{\rho}\left(s, m_1, m_2, \sigma_{\text{noise}}^2, w\right) = w\left(m_1^2\, \sigma_{\text{signal}}^2 + \sigma_{\text{noise}}^2\right). \tag{33}$$

**Lemma A.1.** *If the weight of energetic cost ($w$) is scaled up by an arbitrary non-zero factor $k$, and noise variance ($\sigma_{\text{noise}}^2$) is scaled down by $k$ times, then the channel output ($\boldsymbol{\pi}$) and energetic cost per message ($\boldsymbol{\rho}$) remain unchanged if the pre-multiplier is scaled down by $\sqrt{k}$ times and the post-multiplier is scaled up by $\sqrt{k}$ times.*

*Proof.* The notations used is the same as before, except that for the case of scaling we use overline. That is, $\overline{w} = k\, w, \overline{\sigma}_{\text{noise}}^2 = \sigma_{\text{noise}}^2/k$, and $\overline{\eta} \sim \mathcal{N}\left(0, \overline{\sigma}_{\text{noise}}^2\right)$.

Below, we show that the channel output for the cases with and without scaling is the same.

$$\begin{aligned}
\boldsymbol{\pi}\left(s, \overline{m}_1, \overline{m}_2, \overline{\sigma}_{\text{noise}}^2\right) &= \overline{m}_2\left(\overline{m}_1\, s + \overline{\eta}\right) \\
&= \sqrt{k}\, m_2\left(\frac{m_1}{\sqrt{k}}\, s + \overline{\eta}\right) \\
&= m_2\left(m_1\, s + \sqrt{k}\, \overline{\eta}\right) \\
&= m_2\left(m_1\, s + \eta\right) = \boldsymbol{\pi}\left(s, m_1, m_2, \sigma_{\text{noise}}^2\right).
\end{aligned}$$

The energetic cost per message for the cases with and without scaling is the same as shown below,

$$\begin{aligned}
\boldsymbol{\rho}\left(s, \overline{m}_1, \overline{\sigma}_{\text{noise}}^2, \overline{w}\right) &= \overline{w}\left(\overline{m}_1^2\, \sigma_{\text{signal}}^2 + \overline{\sigma}_{\text{noise}}^2\right) \\
&= k\, w\left(\frac{m_1^2}{k}\, \sigma_{\text{signal}}^2 + \frac{\sigma_{\text{noise}}^2}{k}\right) \\
&= w\left(m_1^2\, \sigma_{\text{signal}}^2 + \sigma_{\text{noise}}^2\right) = \boldsymbol{\rho}\left(s, m_1, \sigma_{\text{noise}}^2, w\right).
\end{aligned}$$

$\therefore$ We have,

$$\boldsymbol{\pi}\left(s, \frac{m_1}{\sqrt{k}}, \sqrt{k}\, m_2, \frac{\sigma_{\text{noise}}^2}{k}\right) = \boldsymbol{\pi}\left(s, m_1, m_2, \sigma_{\text{noise}}^2\right) \tag{34}$$

$$\boldsymbol{\rho}\left(s, \frac{m_1}{\sqrt{k}}, \frac{\sigma_{\text{noise}}^2}{k}, k\, w\right) = \boldsymbol{\rho}\left(s, m_1, \sigma_{\text{noise}}^2, w\right) \tag{35}$$

$\square$

The interpretation of the above result is the following. In comparison to the case without scaling, for the case with scaling, the noise variance is lower $\left(\times \frac{1}{k}\right)$ and the weight of energetic cost is higher $\left(\times k\right)$. Since the noise is lower, a lower amplification $\left(\times \frac{1}{\sqrt{k}}\right)$ would suffice to retain the same information as for the case without scaling. But since the weight of energetic cost is higher $\left(\times k\right)$ for the case of scaling, the lower amplification would still cost the same energetic cost as in the case without scaling.

## A.9 Optimal categorical strategy depends on the feedforward/feedback noise costs

**Theorem A.2.** *If the feedback weight of energetic cost is scaled up by an arbitrary non-zero factor $k$, and the feedback noise variance is scaled down by $k$ times, then (1) the optimal total cost remains unchanged, (2) the optimal control gain $L$ scales down by $\sqrt{k}$ times, and the optimal prediction multiplier $D$ scales up by $\sqrt{k}$ times, (3) the optimal feedback energetic cost remains the same, (4) the optimal feedforward energetic cost, optimal inference cost, optimal multipliers $E$, $F$, and $G$ remain unchanged, while the optimal $H$ scales up by $\sqrt{k}$ times, and (5) the optimal categorical strategy remains unchanged.*

*Proof.* We layout the proof by first considering the *case without scaling* where the weight of feedback energetic cost is $w_b$, and noise variance is $\sigma_b^2$. We express the corresponding optimal control gain $L^\star$, optimal prediction multiplier $D^\star$, and optimal total cost $\text{Cost}_{\text{tot}}^\star$ in terms of the notations used in Lemma A.1. We then consider the *case with scaling*, where the weight of energetic cost is $\overline{w}_{\text{b}} = k\, w_b$, and noise variance is $\overline{\sigma}_b^2 = \frac{\sigma_b^2}{k}$. We express the corresponding optimal control gain $\overline{L}^\star$, optimal prediction multiplier $\overline{D}^\star$, and optimal total cost $\overline{\text{Cost}}_{\text{tot}}^\star$ also in terms of the notations used in Lemma A.1. Finally, we prove the theorem by applying Lemma A.1 and deriving the required equality. For example, to prove the optimal total cost remains unchanged, we derive $\overline{\text{Cost}}_{\text{tot}}^\star = \text{Cost}_{\text{tot}}^\star$.

**Without scaling** $(w_b, \sigma_b^2)$: We start by establishing the equivalence between the notations used in Appendix A.8 and that used for the feedback channel (Eqs 3-5, 8). For the feedback channel, the signal of interest $s$ is $\hat{x}^{t-1}$, the pre-multiplier $m_1$ is the control gain $L$, and the post-multiplier $m_2$ is the prediction multiplier $D$. The channel output is then given as $\boldsymbol{\pi}\left(\hat{x}^{t-1}, L, D, \sigma_b^2\right) = D\left(L\,\hat{x}^{t-1} + \eta_b\right)$, and the energetic cost at time $t$ is $\boldsymbol{\rho}\left(\hat{x}^{t-1}, L, \sigma_b^2, w_b\right)$. The residual (Eq 5) and the feedback energetic cost (Eq 8) can then be expressed as

$$\Delta^t = \boldsymbol{\pi}\left(\hat{x}^{t-1}, L, D, \sigma_b^2\right) + E\,o^t$$

$$\text{Cost}_{\text{b}} = \lim_{T \to \infty} \frac{1}{T} \sum_{t=1}^{T} \boldsymbol{\rho}\left(\hat{x}^{t-1}, L, \sigma_b^2, w_{\text{b}}\right).$$

Note that $D$, $L$, $w_{\text{b}}$, and $\sigma_b^2$ are involved in the dynamics and the optimization Eqs 1-8 only through $\boldsymbol{\pi}\left(\hat{x}^{t-1}, L, D, \sigma_b^2\right)$ and $\boldsymbol{\rho}\left(\hat{x}^{t-1}, L, \sigma_b^2, w_{\text{b}}\right)$ as shown above. Therefore, the minimization of total cost for a given $w_{\text{b}}$, and $\sigma_b^2$ with respect to $D$, and $L$ can be written as

$$\text{Cost}_{\text{tot}}^\star = \min_{L,D} \text{Cost}_{\text{tot}}(\boldsymbol{\pi}\left(\hat{x}^{t-1}, L, D, \sigma_b^2\right), \boldsymbol{\rho}\left(\hat{x}^{t-1}, L, \sigma_b^2, w_{\text{b}}\right)) \tag{36}$$

$$L^\star, D^\star = \arg\min_{L,D} \text{Cost}_{\text{tot}}(\boldsymbol{\pi}\left(\hat{x}^{t-1}, L, D, \sigma_b^2\right), \boldsymbol{\rho}\left(\hat{x}^{t-1}, L, \sigma_b^2, w_{\text{b}}\right)), \tag{37}$$

**With scaling** $(\overline{w}_{\text{b}}, \overline{\sigma}_b^2)$: Just as above, for the scaled case we have

$$\overline{\text{Cost}}_{\text{tot}}^\star = \min_{L,D} \text{Cost}_{\text{tot}}(\boldsymbol{\pi}\left(\hat{x}^{t-1}, L, D, \overline{\sigma}_b^2\right), \boldsymbol{\rho}\left(\hat{x}^{t-1}, L, \overline{\sigma}_b^2, \overline{w}_{\text{b}}\right))$$

$$\overline{L}^\star, \overline{D}^\star = \arg\min_{L,D} \text{Cost}_{\text{tot}}(\boldsymbol{\pi}(\hat{x}^{t-1}, L, D, \overline{\sigma}_b^2), \boldsymbol{\rho}(\hat{x}^{t-1}, L, \overline{\sigma}_b^2, \overline{w}_{\text{b}}))$$

**(1)** By applying Eqs 34-35 of Lemma A.1, and Eq 36 from the case without scaling, we show the optimal costs for both the cases are equal.

$$\overline{\text{Cost}}_{\text{tot}}^\star = \min_{L,D} \text{Cost}_{\text{tot}}(\boldsymbol{\pi}(\hat{x}^{t-1}, L, D, \overline{\sigma}_b^2), \boldsymbol{\rho}(\hat{x}^{t-1}, L, \overline{\sigma}_b^2, \overline{w}_{\text{b}}))$$

$$= \min_{L,D} \text{Cost}_{\text{tot}}(\boldsymbol{\pi}(\hat{x}^{t-1}, L, D, \frac{\sigma_b^2}{k}), \boldsymbol{\rho}(\hat{x}^{t-1}, L, \frac{\sigma_b^2}{k}, k\,w_{\text{b}}))$$

$$= \min_{L,D} \text{Cost}_{\text{tot}}(\boldsymbol{\pi}(\hat{x}^{t-1}, \sqrt{k}\,L, \frac{D}{\sqrt{k}}, \sigma_b^2), \boldsymbol{\rho}(\hat{x}^{t-1}, \sqrt{k}\,L, \sigma_b^2, w_{\text{b}}))$$

$$= \min_{L,D} \text{Cost}_{\text{tot}}(\boldsymbol{\pi}(\hat{x}^{t-1}, L, D, \sigma_b^2), \boldsymbol{\rho}(\hat{x}^{t-1}, L, \sigma_b^2, w_{\text{b}}))$$

$$= \text{Cost}_{\text{tot}}^\star$$

**(2)** Similarly, we have

$$\overline{L}^\star, \overline{D}^\star = \arg\min_{L,D} \text{Cost}_{\text{tot}}(\boldsymbol{\pi}\left(\hat{x}^{t-1}, L, D, \overline{\sigma}_b^2\right), \boldsymbol{\rho}\left(\hat{x}^{t-1}, L, \overline{\sigma}_b^2, \overline{w}_{\text{b}}\right))$$

$$= \arg\min_{L,D} \text{Cost}_{\text{tot}}(\boldsymbol{\pi}\left(\hat{x}^{t-1}, L, D, \frac{\sigma_b^2}{k}\right), \boldsymbol{\rho}\left(\hat{x}^{t-1}, L, \frac{\sigma_b^2}{k}, k\,w_{\text{b}}\right))$$

$$= \arg\min_{L,D} \text{Cost}_{\text{tot}}(\boldsymbol{\pi}\left(\hat{x}^{t-1}, \sqrt{k}\,L, \frac{D}{\sqrt{k}}, \sigma_b^2\right), \boldsymbol{\rho}\left(\hat{x}^{t-1}, \sqrt{k}\,L, \sigma_b^2, w_{\text{b}}\right)).$$

Combining above with Eq 37, we get $\overline{L}^\star = \frac{L^\star}{\sqrt{k}}$, and $\overline{D}^\star = \sqrt{k}\,D^\star$. See Fig 7 for the plot obtained using numerical optimization.

**(3)** Let $\overline{\text{Cost}}_{\text{b}}^{\star}$ be the optimal feedback energetic cost for the scaled case. Applying Eq 35 of Lemma A.1, we show that the optimal feedback energetic cost does not change for the scaled case.

$$\overline{\text{Cost}}_{\text{b}}^{\star} = \lim_{T \to \infty} \frac{1}{T} \sum_{t=1}^{T} \rho\left( \hat{x}^{t-1}, \overline{L}^{\star}, \overline{\sigma}_{\text{b}}^{2}, \overline{w}_{\text{b}} \right)$$

$$= \lim_{T \to \infty} \frac{1}{T} \sum_{t=1}^{T} \rho\left( \hat{x}^{t-1}, \frac{L^{\star}}{\sqrt{k}}, \frac{\sigma_{\text{b}}^{2}}{k}, k\, w_{\text{b}} \right)$$

$$= \lim_{T \to \infty} \frac{1}{T} \sum_{t=1}^{T} \rho\left( \hat{x}^{t-1}, L^{\star}, \sigma_{\text{b}}^{2}, w_{\text{b}} \right)$$

$$= \text{Cost}_{\text{b}}^{\star}.$$

**(4)** As in Eq 36, the dynamics and total cost optimization depend on the feedback relevant terms ($D$, $L$, $w_{\text{b}}$, and $\sigma_{\text{b}}^{2}$) only through the feedback channel output and feedback energetic cost. Under optimal solution, we have that both the feedback channel output and the feedback channel energetic cost remain the same for the cases with and without scaling. That is, using Eqs 34-35 of Lemma A.1, we have $\pi\left( \hat{x}^{t-1}, \overline{L}^{\star}, \overline{D}^{\star}, \overline{\sigma}_{b}^{2} \right) = \pi\left( \hat{x}^{t-1}, L^{\star}, D^{\star}, \sigma_{b}^{2} \right)$, and $\rho\left( \hat{x}^{t-1}, \overline{L}^{\star}, \overline{\sigma}_{\text{b}}^{2}, \overline{w}_{\text{b}} \right) = \rho\left( \hat{x}^{t-1}, L^{\star}, \sigma_{\text{b}}^{2}, w_{\text{b}} \right)$. Therefore, the optimization problem and the dynamics for the cases with and without scaling remain to be equivalent. This results in $\overline{E}^{\star} = E^{\star}, \overline{F}^{\star} = F^{\star}, \overline{G}^{\star} = G^{\star}$, and the optimal feedforward energetic cost and inference cost to be unchanged. For the case with scaling, in addition to the changes in the optimal control gain and prediction multiplier shown above ($\overline{L}^{\star} = \frac{L^{\star}}{\sqrt{k}}, \overline{D}^{\star} = \sqrt{k}\, D^{\star}$), we also have $\overline{H}^{\star} = \sqrt{k}\, H^{\star}$. Note that $\overline{H}^{\star} = \sqrt{k}\, H^{\star}$ does not conflict the statement that optimization problem and the dynamics for the cases with and without scaling remain to be equivalent, but in fact is in agreement. This is because, for the dynamics and optimization to be exactly the same, the coefficient of $\hat{x}^{t-1}$ used in updating the estimate in Eq 7 should also be the same. The coefficient of $\hat{x}^{t-1}$ in Eq 7 is $F + H\, L$. So, when the optimal $L$ reduces by $\sqrt{k}$ factor, optimal $H$ has to increase by $\sqrt{k}$ to keep the coefficient unchanged ($\overline{L}^{\star} = \frac{L^{\star}}{\sqrt{k}}, \overline{H}^{\star} = \sqrt{k}\, H^{\star}$).

**(5)** From above, we have $\overline{L}^{\star} = \frac{L^{\star}}{\sqrt{k}}, \overline{D}^{\star} = \sqrt{k}\, D^{\star}, \overline{E}^{\star} = E^{\star}$. Using this we exhaustively show that whichever is the optimal strategy for the case without scaling, the same would be the optimal strategy for the case with scaling.

If for the case of without scaling is predictive coding, then $L^{\star}, D^{\star}, E^{\star} \neq 0$. For non-zero $k$, this would imply $\overline{L}^{\star}, \overline{D}^{\star}, \overline{E}^{\star} \neq 0$, resulting in predictive coding for the case of scaling.

If for the case of without scaling is efficient coding, then $L^{\star}, D^{\star} = 0$, and $E^{\star} \neq 0$. This would imply $\overline{L}^{\star}, \overline{D}^{\star} = 0$, and $\overline{E}^{\star} \neq 0$, resulting in efficient coding for the case of scaling.

If for the case of without scaling is to not send any messages, then $L^{\star}, D^{\star}, E^{\star} = 0$. For non-zero $k$, this would imply $\overline{L}^{\star}, \overline{D}^{\star}, \overline{E}^{\star} \neq 0$, resulting in sending no messages for the case of scaling. $\quad\square$

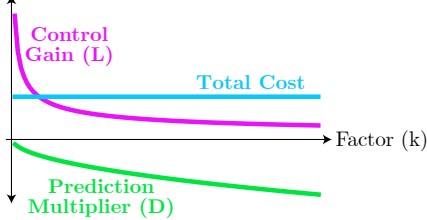

Figure 7: *When the feedback weight of energetic cost is scaled up by $k$ times, and feedback noise variance is scaled down by $k$ times, the optimal costs remain unchanged, and the magnitude of optimal control gain scales down by $\sqrt{k}$ times and the optimal prediction multiplier scales up by $\sqrt{k}$ times. While only the total cost is shown in the figure, we also observe the inference cost, feedback, and feedforward energetic costs to remain the same.*

**Theorem A.3.** *If the feedforward weight of energetic cost is scaled up by an arbitrary non-zero factor $k$, and the feedforward noise variance is scaled down by $k$ times, then (1) the optimal total cost remains unchanged, (2) the optimal observation multiplier $E$ scales down by $\sqrt{k}$ times, and the optimal multiplier for noisy residual $G$ scales up by $\sqrt{k}$ times, (3) the optimal feedforward energetic cost remains the same, (4) the optimal feedback energetic cost, optimal inference cost, optimal multipliers $F$, $H$, $L$ remain unchanged, while the optimal prediction multiplier $D$ scales down by $\sqrt{k}$ times, and (5) the optimal categorical strategy remains unchanged.*

*Proof.* The proof for the feedforward channel is equivalent to the proof for the feedback channel in Theorem A.2, with observation multiplier $E$ as the pre-multiplier $m_1$, and the multiplier for noisy residual $G$ as the post multiplier $m_2$. The theorem is verified using numerical optimization as shown in Fig 8. $\qquad\square$

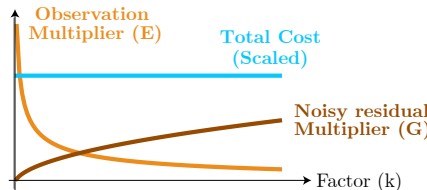

Figure 8: *When the feedforward weight of energetic cost is scaled up by $k$ times, and feedforward noise variance is scaled down by $k$ times, the optimal costs remain unchanged, and the observation multiplier scales down by $\sqrt{k}$ times, and the multiplier for noisy residual scales up by $\sqrt{k}$ times. While only the total cost is shown in the figure, we also observe the inference cost, feedback, and feedforward energetic costs to remain the same.*

**Theorem A.4.** *Under the optimal case, several quantities depend on the feedback weight, feedback noise variance, feedforward weight, and feedforward noise variance only through the feedback noise cost, and the feedforward noise cost. The quantities are (1) the total cost, (2) the optimal categorical strategy, (3) the feedback energetic cost, (4) the feedforward energetic cost, and (5) the inference cost.*

*Proof.* Below, we show the proof for the entity optimal total cost, but the same steps can be followed for proving for other entities listed as well. The layout of the proof is as follows. We consider four different cases of channel noise costs, each denoted by the sequence $(feedback\ weight, feedback\ noise, feedforward\ weight, feedforward\ noise)$. The four cases are $(w_b, \sigma_b^2, w_f, \sigma_f^2)$, $(w_b, \sigma_b^2, \overline{w}_f, \overline{\sigma}_f^2)$, $(\overline{w}_b, \overline{\sigma}_b^2, w_f, \sigma_f^2)$, and $(\overline{w}_b, \overline{\sigma}_b^2, \overline{w}_f, \overline{\sigma}_f^2)$. Where $\overline{w}_b = k_1\ w_b$, $\overline{\sigma}_b^2 = \sigma_b^2/k_1$, $\overline{w}_f = k_2\ w_f$, $\overline{\sigma}_f^2 = \sigma_f^2/k_2$, such that $k_1$ and $k_2$ arbitrary non-zero scaling factors. Since the feedback noise cost is same for $\overline{w}_b\ \overline{\sigma}_b^2 = w_b\ \sigma_b^2$, and feedforward noise cost is same for $\overline{w}_f\ \overline{\sigma}_f^2 = w_f\ \sigma_f^2$, in order to prove the theorem for total cost, it is sufficient to show

$$\text{Cost}_{\text{tot}}^\star\left(w_b, \sigma_b^2, w_f, \sigma_f^2\right) = \text{Cost}_{\text{tot}}^\star\left(w_b, \sigma_b^2, \overline{w}_f, \overline{\sigma}_f^2\right) = \text{Cost}_{\text{tot}}^\star\left(\overline{w}_b, \overline{\sigma}_b^2, w_f, \sigma_f^2\right) = \text{Cost}_{\text{tot}}^\star\left(\overline{w}_b, \overline{\sigma}_b^2, \overline{w}_f, \overline{\sigma}_f^2\right).$$
(38)

From Theorem A.2, we directly get

$$\text{Cost}_{\text{tot}}^\star\left(w_b, \sigma_b^2, w_f, \sigma_f^2\right) = \text{Cost}_{\text{tot}}^\star\left(\overline{w}_b, \overline{\sigma}_b^2, w_f, \sigma_f^2\right) \tag{39}$$

$$\text{Cost}_{\text{tot}}^\star\left(w_b, \sigma_b^2, \overline{w}_f, \overline{\sigma}_f^2\right) = \text{Cost}_{\text{tot}}^\star\left(\overline{w}_b, \overline{\sigma}_b^2, \overline{w}_f, \overline{\sigma}_f^2\right). \tag{40}$$

From Theorem A.3, we get

$$\text{Cost}_{\text{tot}}^\star(w_b, \sigma_b^2, w_f, \sigma_f^2) = \text{Cost}_{\text{tot}}^\star(w_b, \sigma_b^2, \overline{w}_f, \overline{\sigma}_f^2) \tag{41}$$

$$\text{Cost}_{\text{tot}}^\star(\overline{w}_b, \overline{\sigma}_b^2, w_f, \sigma_f^2) = \text{Cost}_{\text{tot}}^\star(\overline{w}_b, \overline{\sigma}_b^2, \overline{w}_f, \overline{\sigma}_f^2). \tag{42}$$

Combining Eqs 39-42, we get Eq 38. Hence proved. $\qquad\square$

## A.10 Channel noise and weight for non-sequential and sequential channels

Below, we derive the results for non-sequential and sequential feedback channels, but the derivation for the feedforward channel is the same. For simplicity, we assume that each neuron has gain $g_{\text{nb}}$, and is corrupted by i.i.d Gaussian noise $\mathcal{N}\left(0, \sigma_{\text{nb}}^2\right)$. Let $w_{\text{nb}}$ be the weight of the energetic cost of

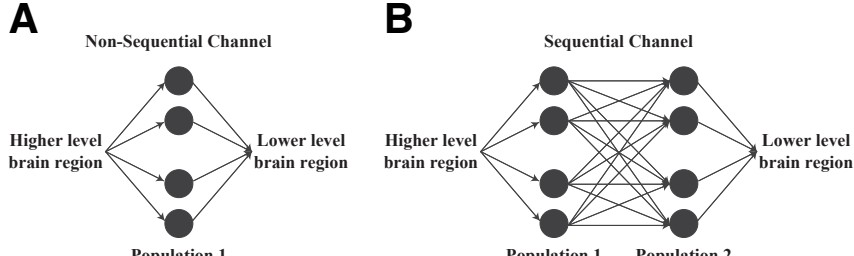

Figure 9: *Example of two different anatomical structures for feedforward/feedback channel. **A**: The non-sequential channel comprises a single population of neurons that linearly encodes the message that needs to be sent. The message received at the end of the channel is the decoding of the population's representation. **B**: The sequential channel comprises two populations of neurons sequentially connected one after the other. The first population linearly encodes the message to be sent, which is then decoded and then encoded by the second population. This is equivalently represented as each neuron in second population being connected to all the neurons in the first population since the weights are all equal. The message received at the end of the channel is a decoding of the second population's representation.*

a single neuron. The subscript nb indicates a single **n**euron in feed**b**ack channel. For the feedback channel, the message to be sent at any time $t$ is the noiseless prediction $p^t$. We compute the equivalent feedback noise variance and weight of the feedback channel in terms of the neuron noise and weight. The computed feedback noise and weight can then be plugged into the original optimization in Eq 8 to find the corresponding solution.

### A.10.1 Non-sequential feedback channel

Let $N_{\mathrm{b}}$ be the number of neurons in the population comprising the feedback channel.

**Feedback channel noise variance:** When the noiseless prediction $p^t$ is linearly encoded by the population of neurons, the decoded signal is $p^t + \eta_{\mathrm{b}}^t$ (as in Eq 4), where $\eta_{\mathrm{b}}^t$ is the feedback noise with feedback noise variance $\frac{\sigma_{\mathrm{nb}}^2}{N_{\mathrm{b}}\, g_{\mathrm{nb}}^2}$.

**Feedback channel energetic cost:** As each of the $N_{\mathrm{b}}$ neurons in the population encode $p^t$ with gain $g_{\mathrm{nb}}$, noise variance $\sigma_{\mathrm{nb}}^2$, and weight of energetic cost $w_{\mathrm{nb}}$, the total feed**b**ack **pop**ulation energy $\mathrm{Cost}_{\mathrm{b}}^{\mathrm{pop}}$ consumed is

$$\mathrm{Cost}_{\mathrm{b}}^{\mathrm{pop}} = \lim_{T \to \infty} \frac{1}{T} \sum_{t=1}^{T} N_{\mathrm{b}}\, w_{\mathrm{nb}}\, g_{\mathrm{nb}}^2\, p^{t\,2} + N_{\mathrm{b}}\, w_{\mathrm{nb}}\, \sigma_{\mathrm{nb}}^2$$

Note that in Eq 8, the feedback energetic cost in the original formulation was written as

$$\mathrm{Cost}_{\mathrm{b}} = \lim_{T \to \infty} \frac{1}{T} \sum_{t=1}^{T} w_{\mathrm{b}}\, \tilde{p}^{t\,2}$$

$$= \lim_{T \to \infty} \frac{1}{T} \sum_{t=1}^{T} w_{\mathrm{b}}\, p^{t\,2} + w_{\mathrm{b}}\, \sigma_{\mathrm{b}}^2$$

Therefore, expressing $\mathrm{Cost}_{\mathrm{b}}^{\mathrm{pop}}$ in terms of $\mathrm{Cost}_{\mathrm{b}}$, we have

$$\mathrm{Cost}_{\mathrm{b}}^{\mathrm{pop}} = \mathrm{Cost}_{\mathrm{b}} + (N_{\mathrm{b}} - 1)\, w_{\mathrm{nb}}\, \sigma_{\mathrm{nb}}^2 \tag{43}$$

where the weight of feedback channel energetic cost is $w_{\mathrm{b}} = N_{\mathrm{b}}\, w_{\mathrm{nb}}\, g_{\mathrm{nb}}^2$. Note that now there is an additional term to $\mathrm{Cost}_{\mathrm{b}}$ in Eq 43 when computing the total feedback energy, and therefore the total cost would also increase by the same amount in Eq 8. However, the optimization in Eq 9 still remains to be exactly the same because the additional term does not include any of the parameters we are optimizing over. Hence, we can solve for the case of non-sequential feedback channel by following the exact same procedure in Section 4.

**Feedback channel noise cost:** We computed the feedback noise variance as $\sigma_{\mathrm{b}}^2 = \frac{\sigma_{\mathrm{nb}}^2}{N_{\mathrm{b}}\, g_{\mathrm{nb}}^2}$, and the feedback weight of energetic cost as $w_{\mathrm{b}} = N_{\mathrm{b}}\, w_{\mathrm{nb}}\, g_{\mathrm{nb}}^2$. Therefore, their product, the feedback

channel noise cost is $w_{\mathrm{nb}}\,\sigma_{\mathrm{nb}}^2$. Note that the feedback channel noise cost is constant for a given noise variance and weight of energetic cost per neuron.

### A.10.2 Sequential feedback channel

Let $\frac{\gamma}{1+\gamma}N_{\mathrm{b}}$, and $\frac{1}{1+\gamma}N_{\mathrm{b}}$ be the number of neurons in the first and second population of the feedback channel. Where $N_{\mathrm{b}}$ is the total number of neurons in both the populations together, and $\gamma$ is the population size ratio. Population size ratio is the ratio between the number of neurons in the first population to the number of neurons in the second population.

**Feedback channel noise variance:** When the noiseless prediction $p^t$ is linearly encoded by the first population of neurons, $p^t + \eta_{1\mathrm{b}}^t$ would be the corresponding linearly decoded signal, where $\eta_{1\mathrm{b}}^t \sim \mathcal{N}\left(0,\ \frac{(1+\gamma)}{\gamma}\frac{\sigma_{\mathrm{nb}}^2}{N_{\mathrm{b}}\,g_{\mathrm{nb}}^2}\right)$. Then, when $p^t + \eta_{1\mathrm{b}}^t$ is linearly encoded by the second population of neurons, the corresponding decoded signal would be $p^t + \eta_{1\mathrm{b}}^t + \eta_{2\mathrm{b}}^t$, where $\eta_{2\mathrm{b}}^t \sim \mathcal{N}\left(0,\ \frac{(1+\gamma)\,\sigma_{\mathrm{nb}}^2}{N_{\mathrm{b}}\,g_{\mathrm{nb}}^2}\right)$. Expressing this noisy prediction in the form of Eq 4, we get $\eta_{\mathrm{b}}^t = \eta_{1\mathrm{b}}^t + \eta_{2\mathrm{b}}^t$, and $\sigma_{\mathrm{b}}^2 = \frac{(1+\gamma)^2}{\gamma}\frac{\sigma_{\mathrm{nb}}^2}{N_{\mathrm{b}}\,g_{\mathrm{nb}}^2}$ as the feedback channel noise variance.

**Feedback channel energetic cost:** As each of the $\frac{\gamma}{1+\gamma}N_{\mathrm{b}}$ number of neurons in the first population encode $p^t$ with gain $g_{\mathrm{nb}}$, noise variance $\sigma_{\mathrm{nb}}^2$, and weight of energetic cost $w_{\mathrm{nb}}$, the energy consumed by the first population of neurons is

$$\lim_{T\to\infty}\frac{1}{T}\sum_{t=1}^{T}\frac{\gamma\,N_{\mathrm{b}}\,w_{\mathrm{nb}}\,g_{\mathrm{nb}}^2}{1+\gamma}\,p^{t2} + \frac{\gamma\,N_{\mathrm{b}}\,w_{\mathrm{nb}}\,\sigma_{\mathrm{nb}}^2}{1+\gamma} \tag{44}$$

And as each of the $\frac{1}{1+\gamma}N_{\mathrm{b}}$ neurons in the second population encode $p^t + \eta_{1\mathrm{b}}^t$, the energy consumed by the second population of neurons is

$$\lim_{T\to\infty}\frac{1}{T}\sum_{t=1}^{T}\frac{N_{\mathrm{b}}\,w_{\mathrm{nb}}\,g_{\mathrm{nb}}^2}{1+\gamma}\,\left(p^t + \eta_{1\mathrm{b}}\right)^2 + \frac{N_{\mathrm{b}}\,w_{\mathrm{nb}}\,\sigma_{\mathrm{nb}}^2}{1+\gamma}$$

$$= \lim_{T\to\infty}\frac{1}{T}\sum_{t=1}^{T}\frac{N_{\mathrm{b}}\,w_{\mathrm{nb}}\,g_{\mathrm{nb}}^2}{1+\gamma}\,p^{t2} + \left(\frac{1}{\gamma} + \frac{N_{\mathrm{b}}}{1+\gamma}\right)w_{\mathrm{nb}}\,\sigma_{\mathrm{nb}}^2 \tag{45}$$

Combining energy of both the populations (adding Eqs 44 and 45), we get total feedback **pop**ulation energy $\mathrm{Cost}_{\mathrm{b}}^{\mathrm{pop}}$ as

$$\mathrm{Cost}_{\mathrm{b}}^{\mathrm{pop}} = \lim_{T\to\infty}\frac{1}{T}\sum_{t=1}^{T}N_{\mathrm{b}}\,w_{\mathrm{nb}}\,g_{\mathrm{nb}}^2\,p^{t2} + \left(N_{\mathrm{b}} + \frac{1}{\gamma}\right)w_{\mathrm{nb}}\,\sigma_{\mathrm{nb}}^2$$

Note that in Eq 8, the feedback energetic cost in the original formulation was written as

$$\mathrm{Cost}_{\mathrm{b}} = \lim_{T\to\infty}\frac{1}{T}\sum_{t=1}^{T}w_{\mathrm{b}}\,\tilde{p}^{t2}$$

$$= \lim_{T\to\infty}\frac{1}{T}\sum_{t=1}^{T}w_{\mathrm{b}}\,p^{t2} + w_{\mathrm{b}}\,\sigma_{\mathrm{b}}^2$$

Therefore, expressing $\mathrm{Cost}_{\mathrm{b}}^{\mathrm{pop}}$ in terms of $\mathrm{Cost}_{\mathrm{b}}$, we have

$$\mathrm{Cost}_{\mathrm{b}}^{\mathrm{pop}} = \mathrm{Cost}_{\mathrm{b}} + \left(N_{\mathrm{b}} - \gamma - 2\right)w_{\mathrm{nb}}\,\sigma_{\mathrm{nb}}^2 \tag{46}$$

where the weight of feedback channel energetic cost is $w_{\mathrm{b}} = N_{\mathrm{b}}\,w_{\mathrm{nb}}\,g_{\mathrm{nb}}^2$. Note that now there is an additional term to $\mathrm{Cost}_{\mathrm{b}}$ in Eq 46 when computing the total feedback energy, and therefore the total cost would also increase by the same amount in Eq 8. However, the optimization in Eq 9 still remains to be exactly the same because the additional term does not include any of the parameters we are optimizing over. Hence, we can solve for the case of sequential feedback channel by following the exact same procedure in Section 4.

**Feedback channel noise cost:** We computed the feedback noise variance as $\sigma_{\mathrm{b}}^2 = \frac{(1+\gamma)^2}{\gamma} \frac{\sigma_{\mathrm{nb}}^2}{N_{\mathrm{b}} \, g_{\mathrm{nb}}^2}$, and the feedback weight of energetic cost as $w_{\mathrm{b}} = N_{\mathrm{b}} \, w_{\mathrm{nb}} \, g_{\mathrm{nb}}^2$. Therefore, their product, the feedback channel noise cost is $\frac{(1+\gamma)^2}{\gamma} \, w_{\mathrm{nb}} \, \sigma_{\mathrm{nb}}^2$. Note that the feedback channel noise cost does not depend on the number of neurons $N_{\mathrm{b}}$, but depends on the ratio $\gamma$ that determines the anatomical structure.