# OpenReview forum: "Phase transitions in when feedback is useful"
_NeurIPS.cc/2022/Conference — NeurIPS 2022 Accept_

### Official Review · Reviewer_v9qs · 2022-07-06

**Rating:** 7
**Confidence:** 3
**Soundness:** 4 excellent
**Presentation:** 3 good
**Contribution:** 3 good

**Summary:**

This paper inspects the consequences of considering the costs associated with sending messages to represent univariate variables in the brain. The authors propose a dynamical system taking into account the inherent noise while updating the estimation of the latent world state. They subsequently derive a theory from the minimization (supposed to take place in the brain) of a cost function taking into account the accuracy of the latent state estimation as well as the feedforward and feedback costs. The main part of the paper presents a mathematical analysis of the system, that is, shows phase transitions between predictive coding (feedforward + feedback messages), efficient coding (feedforward messages only) and silence (no messages), depending on the values of parameters. Finally, the theoretical framework is put into an experimental perspective, through parallels with the properties of the biological brain and discussion about how to test the theoretical predictions.

**Questions:**

1. Is it "just" a model of the brain, or could it be useful in another domain? I feel like this theory could be used in designing energy-efficient systems where the estimation takes place under constraints (for instance energetic). This could be acknowledged in the discussion
2. How could the abstract message-passing in the graph from Fig 1B relate to the messages sent from excitatory/inhibitory populations?
3. Mathematical conventions: to relate more to predictive coding, shouldn't L be positive and D negative, instead of the reverse? Potentially also replace D with -D, in order to have a minus sign in Eq 5

----------
Comments:
1. I think Fig 9 should be moved from the appendix to the main text
2. I didn't understand much Fig 4 with the legend. For instance, which color represents the absence of messages? None, right?

**Limitations:**

Limitations: The authors carefully address the limitations of the work, opening the way for generalizations of the theory (taking actions / multivariate variables / nonlinear tasks / learning).

Societal impact: The authors did not address any potential negative societal impact of their work, which is fine given the subject of the paper (understanding the brain)

**Strengths And Weaknesses:**

Strengths:
1. Interesting topic, well in the scope of the NeurIPS conference
2. Very pedagogical figures and legends
3. The paper has a very clear and intuitive message
4. The paper is well-grounded mathematically
5. The link from the very theoretical framework to the biological brain is very interesting. More particularly, I enjoyed the parts about the excitatory/inhibitory processes in the brain, especially the explanation about the potential higher cost of long-range inhibitory feedback as a consequence of the anatomical constraints.

Weaknesses:
1. The authors put some effort into thinking of ways to test their theory, but the discussion remains a bit too conceptual and vague, especially the part about the comparison between brain systems or species. Add references and explicit further your predictions
2. The relation with previous theoretical work is only superficial (see section "Related work'). I would expect more developments in the discussion part, in light of the paper's results

---

> ### Author Response · Authors · 2022-08-02
> **better predictions and comparisons**
>
> We are happy that the reviewer found the paper interesting, largely clear, well-grounded, and biologically relevant.
>
> Weakness 1: We think that Figure 6 reflects a clear set of predictions for response properties under experimentally controllable conditions.
> However, we agree that greater specificity is warranted for predictions about different architectural constraints. Fundamentally this is about finding biological correlates of the feedforward and feedback noise costs. This has been the most difficult kind of prediction to connect to biology, as those costs depend on many details of the architecture, many of which have not been characterized experimentally with sufficient detail. Nonetheless, we now see a simpler way to state these predictions: instead of speculating about these costs we can propose measuring the noise and projective field (a proxy for the sensitivity of the total activity to a small change in input, ie the metabolic cost of a signal), and the product of these is an estimate of the noise cost. We then predict that brain areas or animals with highest and lowest feedforward costs will use the least feedback (as in Fig 4).
>
> Weakness 2: Due to space constraints, we previously highlighted only the fundamental differences from previous work, namely that they all place limitations on information content and not on computation. However, we agree that it will be helpful to revisit these works in the discussion. In particular, we will add:
>
> Past work described temporal decorrelation as optimal encoding strategy, subtracting a slow, predictable component from the recent measured signal, implementing a kind of predictive coding [Srinivasan 1982, van Hateren 1993]. Like our results, they too observed that this strategy changes with the signal-to-noise ratio (SNR). Efficient coding shows that mean responses are fully subtracted at all SNRs, but with different temporal kernels [Srinivasan 1982, van Hateren 1993]. Later studies of predictive coding observed that for low SNR, temporal decorrelation is no longer the best strategy [Chalk et al 2018]. Our work generalizes these results to account not only for feedforward noise and costs in sensory coding, but also for noise and costs in recurrent computation. In particular, the nonmonotonic dependence of optimal feedback on feedforward noise costs is a novel consequence of our framework with no analog in previous studies.
>
> We will also discuss how adaptive strategies, like [Młynarski and Hermundstad], could implement our predicted changes in the optimal suppression as sensory statistics change. Conversely, by considering the computational costs of adaptation itself (and not just the activity resultant from adapted computation), we may find conditions where it is not beneficial to adapt.
>
> Q1: As pointed out by the reviewer, the model could indeed be useful in other domains as well. For example, low-power applications of inference and control, such as drones or remote sensing, often place constraints on dynamic range and noise (e.g. quantization noise) for all signals, and could thus benefit from optimizations based on our model. This will be acknowledged in the discussion section.
>
> Q2: The feedforward pathway (the arrow from node representing ‘delta’), and the feedback pathway (the arrow from node representing ‘p’), are biologically implemented using activations of excitatory/inhibitory neuronal populations.
>
> Q3: We agree with the suggested sign convention, and will adopt it for the next version.
>
> Comment 1: we will incorporate a small version of Fig 9 in the main paper.
> Comment 2: Fig 4 will be expanded to incorporate the region of no messages, and color-coded accordingly.

---

> > ### Comment · Reviewer_v9qs · 2022-08-08
> > **Response**
> >
> > I am happy with the answers provided by the authors, especially about the plan to answer Weakness 2. I remain positive about this paper, as shown by my (unchanged) evaluation.
> >
> > Note: I modified in Q1 the incomplete sentence "where the estimation is." to "where the estimation takes place under constraints (for instance energetic)".

---

### Official Review · Reviewer_15Ei · 2022-07-11

**Rating:** 7
**Confidence:** 3
**Soundness:** 4 excellent
**Presentation:** 3 good
**Contribution:** 3 good

**Summary:**

This paper formulates an interesting tradeoff in optimal coding theory, in the context of (potential) predictive coding where there are feedforward messages, and top-down predictions can be sent as feedback to reduce feedforward signal cost. Previous works considered the cost of feedforward information but ignored the cost of feedback computation; this paper considers the noise cost of the feedback pathway, as well as the feedforward, and investigates the tradeoff as the noise costs in the feedforward/feedback channels vary. Interestingly, the paper shows that there is a nontrivial transition between three regimes: where predictive coding is preferred (top-down prediction feedback is useful), where efficient coding is preferred (feedback is not worth the cost, so just send feedforward signals), and where sending any messages is not worth the cost at all. The paper also provides biological evidences to argue that it is indeed plausible to have different noise costs in the feedforward/feedback pathways.


**Questions:**

i) The theory is developed in the simple LQG model setting. Can you say whether the results in this limit corresponds to any bounds to the non-linear, non-gaussian cases?

ii) One of the most interesting points in this paper is that the non-monotonic shape of the boundary curve $\Phi$ as a function of the feedforward noise cost (Eq 14, Fig 3). And as soon as I saw it, I was eager to understand why it was so. When $ \Phi $ first appears in the paper, the authors only say that it is a "complicated function", but it would be nice to provide some more insights on why it increases at first and decreases later. Looking at Appendix A7 (as referred to from Eq 14) did not help much. I eventually found relevant text later in the paper, in section 5.2 (paragraphs starting L232 and L242). I suggest that this material be moved up, so that the reader gets a clearer picture early on. It was nice that the authors explain it with a fixed, low-to-moderate feedback noise cost, and talk about small / intermediate / large feedforward noise regimes. It would help to add a horizontal line to Fig 3A and describe it right there.

iii) Minor comments:

- I felt that the text was rather dense in my initial read, and the main messages were not easily located by quick skimming. Hard to pinpoint a specific suggestion, but in general, having the main point at the beginning of each paragraph is really helpful.
- In the figures, red-green-yellow will make it difficult for color-blind readers. Suggest replacing with other color palettes.
- The 3-d plot in Fig 4 left was difficult to understand at first, because the contour lines wasn't visible in the normal pdf view. (I realized after zooming in that there were contour lines.)


**Limitations:**

The authors have addressed the limitations of their modeling assumptions and constraints.

**Strengths And Weaknesses:**

This is an important and interesting paper that studies a fundamental tradeoff in the theory of neural computations. I believe that this paper provides a conceptual advance and will be a significant contribution to the community's understanding of optimal coding. The neurobiological discussion at the end is valuable as well. The paper is original in their consideration of the feedback noise cost, and the discovery of the nontrivial structure of the tradeoff.

I find the mathematical formulation very reasonable and the logical developments convincing, although I have not checked all steps carefully.

As for the weaknesses of the study, the theory is formulated in a simplified model setting, and it is still to be tested how relevant the lessons are in the actual neural systems. Also, this paper provides no direct connection to real neural data. But this is how theories should start, and the authors also discuss the limitations in the paper, with plans for future work.

Presentation-wise, the paper is nicely contextualized with respect to the existing ideas and models. Regarding writing/visualization, I eventually felt that the paper is clear enough, in that it describes all important points, and touches on most of my questions that came up over the course of reading. That said, in the initial read I had to go back and forth between the figures and the text paragraphs more than I would usually do. I will provide more comments below.

---

> ### Author Response · Authors · 2022-08-02
> **more clarity and earlier explanations of nonmonotic behavior**
>
> We are glad to hear the reviewer found many positive qualities in our study.
>
> Question i) While we do not have any results for the nonlinear, non-Gaussian case, we will state our prediction that the core intuition will persist: feedback should not be used when it is too noisy or expensive, or when feedforward messages are too expensive. It is likely that more complicated phase transitions could emerge as a result of increased nonlinearities and their interactions with noise.
>
> Question ii) We thank the reviewer both for suggesting an earlier explanation of the non-monotonic behavior, and for suggesting a good placement of this explanation. We will introduce the text from L232 earlier as suggested, and will add a horizontal line to Fig 3A as a visual guide for the explanation.
>
> Question iii) For the revision we will apply more pressure to simplify and clarify. We will replace red-yellow-green (chosen for the traffic light connotations for feedback) with a red-gray-cyan color scheme. We will increase the line width of the contours.

---

> > ### Comment · Reviewer_15Ei · 2022-08-06
> > **response to authors**
> >
> > I read the authors' responses and the revision plan sounds good to me. Under the assumption that the authors will incorporate the feedback as promised and work to improve clarity, I remain enthusiastic about this paper.

---

### Official Review · Reviewer_8Atc · 2022-07-11

**Rating:** 8
**Confidence:** 4
**Soundness:** 4 excellent
**Presentation:** 3 good
**Contribution:** 4 excellent

**Summary:**

This paper presents a theory of optimal inference in presence of noisy
sensation, prediction and feedback, under linear-quadratic-gaussian
assumptions, with separate costs for feedforward and feedback
signals. The theory is used to show how the usefulness of feedback
undergoes phase transitions depending on the structure of the
environment, architectural constraints, and computational costs.


**Questions:**

Here are some passages that I found harder to read than necessary:
- Line 182: I was confused by the sentence "Furthermore, we also see
  that when sending feedback is optimal, it must be worth sending some
  feedforward messages even if feedback is cut off". In that context,
  it is not clear whether the reader is supposed to "see" that fact
  from some of the equations above, or this is supposed to be
  self-evident based on the consideration made in the rest of the
  sentence. Moreover, I was surprised by the use of "cut off" -
  somehow it made me doubt that I had missed a component of the model
  that accounted for specific causal interventions on the system.
- Lines 206--209, series of sentences starting with "At the
  origin...": this same idea was much clearer when repeated below,
  around lines 235--238 (even if for a slightly different case). In my
  opinion, adopting the phrasing of 235 the first time that the
  concept is presented would make for smoother reading.
- Unless I'm missing something, lines 262--265 seem redundant with
  266--270. The two passages follow immediately one another, but they
  are in different paragraphs. Please consider merging these passages
  if they are indeed redundant, or signposting better their difference
  if they are not.


**Limitations:**

In my opinion, the paper gives a fair account of its limitations.


**Strengths And Weaknesses:**

### Strengths
This paper tackles an important problem in theoretical
neuroscience, hitting a sweet spot between level of detail and
tractability of the system under exam. The method and the results
provide meaningful progress towards understanding efficient coding in
dynamical contexts and its relationship with predictive coding, a
research area that has seen intense effort and several high-quality
contributions in the last few years (such as those by Chalk & Tkacik
and Mlynarski & Hermundstad, which the paper prominently acknowledges).

### Weaknesses
In the central part of the paper (sections 5.1, 5.2), some
passages could be made clearer/easier to follow. Please see
"questions" for details.

---

> ### Author Response · Authors · 2022-08-02
> **improving clarity**
>
> We are glad the reviewer finds our advances novel and important, and appreciates how we balance tractability and salient biological structure. We thank the reviewer for their careful reading and their detailed suggestions about how to improve our paper.
>
> Line 182: The next sentence explains why feedback implies useful feedforward signals. However, we have encountered similar confusion when reading text where the subsequent explanation is not adequately foreshadowed. Thus we will reorder the sentences so that the reader is not left wondering.
>
> We will adopt the phrasing of line 235 to clarify lines 206–209.
>
> Lines 262–5 and 266–70 are intended to convey different messages: the first describes the transition of feedforward messages when feedback is too costly; the second describes the transition of feedback as feedback costs vary. We see that this is confusing and will reword for greater clarity.

---

> > ### Comment · Reviewer_8Atc · 2022-08-07
> > **Response to authors**
> >
> > I thank the authors for taking note of my suggestions for improving the clarity of the paper.

---

### Official Review · Reviewer_2STF · 2022-07-11

**Rating:** 5
**Confidence:** 3
**Soundness:** 2 fair
**Presentation:** 2 fair
**Contribution:** 2 fair

**Summary:**

The present paper studies the inference of a hidden Markov model, specifically a Kalman filter-like problem. Compared with a common Kalman filter and canonical predictive coding framework, the paper considers that the feedback input (prediction) is associated with a computational cost, and how the feedback cost affects the gain of observation and predictions.

**Questions:**

Please see my other comments.

**Ethics Review Area:**

["I don’t know"]

**Limitations:**

A strong requirement of a network implementation of the Kalman filter is that a network model with fixed parameters is able to infer the latent variables with different uncertainties, e.g., different observation noise or feedback noise. Not sure whether the current model could achieve this or not. It might be good to discuss this point.

**Strengths And Weaknesses:**

The formulation of the problem by considering a cost associated with feedback is novel, and the whole paper is structure-wise, well-written. Nevertheless, considering the neural implementation of the Kalman filter has been widely studied before, the formulation of the paper seems a bit technically simplified, and it is not clear whether the novelty of significance of the present paper is enough even though the consideration of feedback cost is new. Also, the paper directly deals with the Gaussian variables (Eq. 1-7) while we know the neural variability is Poisson-like (multiplicative). It is not clear when considering the multiplicative noise whether the main conclusions of the paper can be held. Moreover, the previous studies of neural implementation of Kalman filters typically embed the Gaussian dynamics (Eqs. 1-7) into nonlinear, biologically plausible network dynamics (e.g., Deneve, J. Neurosci., 2007; Wilson and Finkel, NeurIPS 2009; Kutschireiter 2015), hence it might be more interesting to study how the feedback cost influences the network dynamics.

---

> ### Author Response · Authors · 2022-08-02
> **our simplifications are helpful, even though they are not the final word**
>
> We wholeheartedly agree that generalization is an important direction for future work. We are currently pursuing variants with nonlinear dynamics, non-gaussian and non-additive noise, structured multidimensional variables, and sparse network implementations. We expect interesting and distinctive phenomena from some of these extensions, as well as many commonalities with the linear system.
>
> However, the current framework already provides foundational concepts and consequences, shedding light on the existence of non-trivial phase transitions even for linear gaussian systems with quadratic costs. While a more accurate assessment of costs could indeed come from detailed network mechanisms, especially from metabolic costs in spiking networks, these costs can be summarized and approximated to yield more tractable insights. There is a rich and influential tradition of identifying and evaluating new principles with simplified systems — even specifically with linear Gaussian dynamics. Consider the Gaussian channel (Shannon 1948), the Kalman filter (Kalman 1960a), LQR control (Kalman 1960b), efficient coding (Atick and Redlich 1990, van Hateren 1990, Doi and Simoncelli 2012), Gaussian Belief Propagation (Weiss and Freeman 2001), the Gaussian Information Bottleneck (Chechik et al 2001), and even data analyses based on Granger causality or dynamic causal models which assume linear models with Gaussian noise: all of these provide important insights through simplification. These models are all wrong, but useful [Box].
>
> Even with the extensions mentioned above, we expect that some of our core intuitions should hold: feedback messages should be avoided when either feedforward or feedback is expensive and noisy. However, demonstrating this in more realistic settings presents considerable difficulties that we do not think should preclude publication of our foundational results.
>
>
> Limitations: We agree that appropriate adaptation to uncertainty is a key property of a Kalman filter, and one not possessed by a fixed, nonadaptive linear integrator. For a fixed network to implement distinct Kalman filters for distinct input statistics (e.g. different Kalman gains / integration time constants for different signal-to-noise ratios), one needs a nonlinear system that appropriately maps those input statistics onto local integration properties (e.g. Kalman gain). This is an important but challenging adaptation problem for any network implementation of probabilistic inference that is valid under changing uncertainty [e.g. Rao 2010; Beck, Lathem, Pouget 2011; Raju and Pitkow 2016]. However, since our work describes the phenomenological consequences of such filtering, rather than a network mechanism, such adaptation should be beyond the scope of this paper.

---

### Meta-Review · Area_Chair_NpbP · 2022-08-26

**Recommendation:** Accept
**Confidence:** Certain

**Metareview:**

Sound and clearly presented contribution to the field of predictive coding.

**Award:**

No

---

### Decision · Program_Chairs · 2022-09-14

Accept